# Microbial methane cycling in a landfill on a decadal time scale

Daniel S. Grégoire [1,2] ✉, Nikhil A. George [1] & Laura A. Hug[1] ✉

Landfills generate outsized environmental footprints due to microbial degradation of organic matter in municipal solid waste, which produces the potent greenhouse gas methane. With global solid waste production predicted to increase substantially in the next few decades, there is a pressing need to better understand the temporal dynamics of biogeochemical processes that control methane cycling in landfills. Here, we use metagenomic approaches to characterize microbial methane cycling in waste that was landfilled over 39 years. Our analyses indicate that newer waste supports more diverse communities with similar composition compared to older waste, which contains lower diversity and more varied communities. Older waste contains primarily autotrophic organisms with versatile redox metabolisms, whereas newer waste is dominated by anaerobic fermenters. Methane-producing microbes are more abundant, diverse, and metabolically versatile in new waste compared to old waste. Our findings indicate that predictive models for methane emission in landfills overlook methane oxidation in the absence of oxygen, as well as certain microbial lineages that can potentially contribute to methane sinks in diverse habitats.

Landfills worldwide are one of the key mitigation gaps in managing global methane emissions. From 2000 to 2017, landfills produced 60 to 69 Tg of methane per year[1,2]. In high GDP per capita countries (i.e., defined by the World Bank as countries with a GDP per capita >49,000 USD in 2022) such as the United States (GDP per capita 76,389 USD in 2022)[3], landfills can account for up to 20% of net methane emissions[4]. As of 2018, it is estimated that 2.01 billion tonnes (i.e., 2010 Tg) of solid waste has been produced globally, with 35% of this waste being landfilled[5]. With solid waste production predicted to increase to 3.40 billion tonnes (i.e., 3400 Tg) by 2050[5], there is a pressing need to better understand the biogeochemical processes that control methane's fate in landfills to enable the development of waste management practices that mitigate greenhouse gas (GHG) emissions.

Landfilled waste is spatially and geochemically heterogeneous, with compositional changes occurring over extended time scales. Part of the challenge in managing GHG emissions from municipal solid waste (MSW) lies in predicting how these variables interact to affect methane cycling over decades of landfill operation.

The major biogeochemical transitions that occur in a sanitary landfill can be summarized using a five-phase conceptual model [[6,7] and references therein]. Phase 1 is the aerobic phase, where chemoheterotrophic microbes consume oxygen to metabolize organic carbon from paper, food waste, and cover soils. Phase 2 is the anaerobic acid phase, where fermentative microbes hydrolyze cellulose-bearing waste and produce labile organic substrates that support fermentation and organic acid production, which decreases pH in the landfill. Phase 3 is characterized by rapid methanogenesis, where labile organic and inorganic carbon substrates stimulate biogenic methane production by anaerobic archaea. Phase 4 is delineated by a transition to slow methanogenesis, where substrates that support methanogenesis have been depleted and methane production slows. In phase 5, oxygen infiltration can occur because substrates that support aerobic heterotrophy have been exhausted, such that aerobic respiration cannot outpace oxygen diffusion from the atmosphere. Phase 5 is considered the point at which MSW has stabilized yet remains the least well-understood of the lifecycle phases

[1]Department of Biology, University of Waterloo, Waterloo, ON N2L 3G1, Canada. [2]Present address: Department of Chemistry, Carleton University, Ottawa, ON K1S 5B6, Canada. ✉e-mail: danielgregoire@cunet.carleton.ca; laura.hug@uwaterloo.ca

because it can take over 20 years to develop and there are limited datasets covering this timespan.

Microbial metabolisms drive every major biogeochemical change that occurs in a landfill. A key 16S rRNA amplicon sequencing survey that sampled 19 landfills in the United States suggested that the age of refuse and local environmental conditions play significant roles in shaping microbial communities in landfill leachate[8]. Several smaller-scale amplicon sequencing studies have suggested that variables including age[9], nutrient concentrations[10], physicochemical parameters (e.g., temperature and pH)[11], and contaminant concentrations[12–14] all shape microbial community structure in landfills. 16S rRNA surveys have also been used to characterize the succession of microbial taxa over the course of waste degradation in controlled settings[15–18]. Methane cycling guilds in landfills have been examined using 16S rRNA primers specific to methanogens and methanotrophs[19,20]. Sequencing the *mcrA* gene, which codes for the methyl coenzyme M reductase responsible for converting methyl-coenzyme M to methane during methanogenesis[21], has expanded the diversity of methanogenic taxa associated with landfills[22,23]. Similarly, sequencing the *pmoA* gene, which codes for a key subunit of the particulate methane monooxygenase (pMMO), and the *mmoX* gene, which codes for a key subunit in the soluble methane monooxygenase (sMMO), has clarified the structure of methanotrophic communities in landfill cover soils[24–26].

In the case of methanogens, taxonomy determined via amplicon sequencing data is routinely used to infer whether hydrogenotrophic (i.e., $H_2$ and $CO_2$ requiring) or acetoclastic (i.e., acetate requiring) methanogenesis contributes to methane production. Taxonomy is also used to determine whether methane oxidation is carried out by bacteria that require low levels of methane and high levels of key trace nutrients (i.e., Type I methanotrophs) or high levels of methane but low levels of other key substrates (i.e., type II methanotrophs)[27]. These approaches are limited when applied to novel methane cycling taxa that are not related to well-characterized model organisms and tend to ignore more diverse methane cycling metabolisms that cannot be captured by the dichotomies indicated above[28]. This includes intra-aerobic and anaerobic methane oxidation metabolisms that do not require exogenous oxygen, which is rarely considered despite landfills being dominated by anoxic habitats conducive to these lifestyles.

The recent application of metagenomic sequencing to landfills offers a promising solution to the limitations of previous work. Metagenomics has allowed valuable insights into the physiological pathways contributing to waste degradation including cellulose metabolism[29,30] and plastic biodegradation[31]. Metagenomics has also been used to address human health concerns tied to landfills such as the occurrence of antibiotic resistance in pathogens found in MSW[32,33]. Genome-resolved metagenomics[34] can be used to characterize microbial diversity across physicochemical gradients, including identifying factors constraining the distribution of methanogens in MSW[35] and the range of methanotrophic lifestyles that can be supported (e.g., facultative vs. obligate, aerobic vs. microaerophilic vs. anaerobic). Genome-resolved metagenomics also provides the opportunity to consider the contributions of methane-cycling organisms in the broader context of nutrient-cycling pathways and central redox metabolisms that control landfill biogeochemistry[36]. Although amplicon sequencing surveys have shed light on the changes in community structure that occur across landfill habitats, metagenomic surveys that examine the major guilds and physiological pathways controlling biogeochemical cycles over the spatial and temporal scales relevant to landfill lifecycles are lacking.

Here, we use metagenomic sequencing to provide a historical perspective on microbial communities, with an emphasis on methane cycling guilds, in a sanitary landfill across time. In this study, we compare methanogen and methanotroph community structure and metabolic capacity in leachate samples from a landfill spanning the five landfill lifecycle phases. We use phylogenomic analyses and metabolic

models to identify adaptations in methanotrophs and expand the diversity of taxa potentially capable of oxidizing methane in oxygen-limited landfill habitats. Finally, we demonstrate how the biodiversity in landfills includes and allows the identification of microbes with uncharacterized methane cycling capabilities whose role in the global methane cycle has been overlooked.

## Results and discussion
### Landfill lifecycle geochemistry
Landfill cells A, B, and C represent older waste. Cell A was filling from 1980-1982 (39 years old), cell B was filling from 1982–1988 (37 years old), and cell C was filling from 1988–1993 (31 years old). Landfill cells A and B can be classified to phase 5 of the landfill lifecycle whereas cell C is in phase 4. These classifications stem from the low gas production observed across all three cells and the evidence of oxygen intruding into cells A and B (Fig. 1). Gas data aligned with leachate geochemistry, which showed that biological and chemical oxygen demand (BOD and COD, respectively) decreased in line with the low availability of organic substrates at these locations (Figs. S1, S2). The weak evidence of oxygen intrusion and the higher BOD/COD ratio recorded in cell C vs. A and B resulted in cell C's classification to phase 4 (Table S1).

Landfill cell D contains two sub-cells, D1 which was filling from 1993-1998 (26 years old), and D2 which was filling from 1995–1998 (24 years old). Both locations in cell D were classified to phase 4 of slow methanogenesis based on the low levels of gas produced despite cell D's comparable size to cells E and F (Fig. 1A and Table S1). The gas at location D2 was largely comprised of methane although the physical

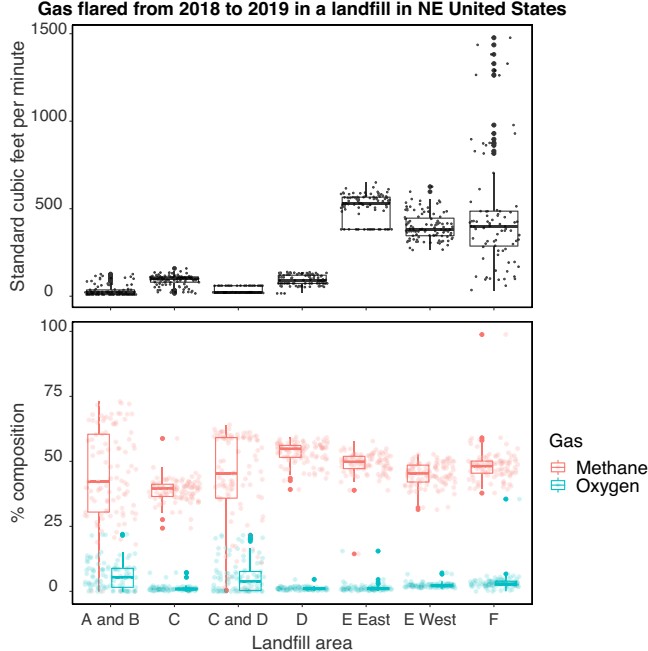

**Fig. 1 | Total gas flared (upper panel) and the relative composition of the gas flared (bottom panel) from vents associated with landfill cells A, B, C, D, E, and F.** The number of observations for total gas flared and gas composition is $n = 103$ for cells A and B, cell C, and cell E West. The number of observations for total gas flared and gas composition is $n = 101$ for cells C and D, cell D, and E East. The number of observations for total gas flared and gas composition is $n = 84$ for cell F. The bottom and the top of the boxes show the first and third quartiles respectively, the bar in the middle shows the median value, whiskers show the minimum and maximum values within 1.5 times the interquartile range, and values that extend beyond 1.5 times the interquartile range are shown as points above and below the whiskers. Data were provided by the site owners for 2018-2019 to cover seasonal variation over two years and capture current trends in gas emissions. Source Data are provided as a Source Data file.

connection between the C/D valley vent and location D1 suggests some oxygen may be intruding at D1 (Fig. 1B). Gas data aligned with aqueous geochemistry wherein peaks in BOD, COD and organic acids have long since passed at both locations (Figs. S1, S2). Location D2 showed slightly higher gas production compared to D1, which may be attributed to the higher BOD/COD ratio. This increased BOD/COD ratio at the time of sampling reflects higher concentrations of carbon sources such as acetate (74 mg L$^{-1}$ vs. 2.3 mg L$^{-1}$) that can support methanogenesis (Figs. 1, S2, and Table S1).

Cell E was filling from 1999–2014 (20 years old) and was classified as being in phase 3 of rapid methanogenesis based on total gas production being almost an order of magnitude higher compared to cells A, B, C, and D and gas being largely comprised of methane (Fig. 1). Cell E reached peak BOD and COD over a similar time frame to cell D and showed circumneutral pH and reducing conditions in line with phase 3 of the landfill lifecycle (Figs. S1, S2). Notably, cell E showed lower BOD/COD compared to D2 suggesting this is not a sufficient standalone index of methanogenic potential (Table S1). Cell E also experienced historically lower peaks in organic acids compared to cells A, B, C, and D (Fig S2). These results could be attributed to a yard waste diversion program implemented in 2000 or a physical connection to landfill cell D designed to redistribute leachate evenly between cells D and E (Table S1). Lower moisture in cell E would have limited the circulation of nutrients and the maintenance of anoxic habitats required for fermentative acid production such that organic acid concentrations decreased. Cell E also experienced a decline in bicarbonate concentrations, reaching concentrations as low as 851 mg L$^{-1}$ before increasing to concentrations comparable to the peaks observed for landfill cells A, B, C, and D (i.e., ~3500 mg L$^{-1}$) (Fig S2). Although environmental conditions seem homogenous enough in cell E to support consistent methane production across two different sampling vents, we make note of the decrease in bicarbonate because it occurred on the day of our sampling trip (2019-12-12) which may have impacted the observed microbial community.

Landfill cell F contains two sub-cells, F1 which was filling from 2014-present (5 years old), and F2 which was filling from 2016 to present (3 years old). Cell F is classified as transitioning from phase 2 of anaerobic acid production to phase 3 of rapid methanogenesis based on recent depletions in organic carbon occurring alongside highly variable amounts of methane production. Cell F displayed a similar gas composition to cell E with gas being largely comprised of methane but subject to highly variable total gas production (i.e., maxima recorded of >1000 cubic feet per minute and minima as low as 35 cubic feet per minute) (Fig. 1). Location F1 experienced an increase in COD and recent decrease in BOD whereas location F2 saw an increase in COD and BOD alongside peaks in organic acids and the highest BOD/COD ratio recorded across the landfill (Figs. S1, S2, and Table S1). These observations suggest that an influx of organic acids that can be oxidized is occurring at cell F, which is further supported by increasing bicarbonate concentrations at both locations (Fig S2). These changes in gas and aqueous geochemistry align with what we would expect of a microbial community where methanogens are competing with heterotrophs for organic substrates as the landfill transitions from a state of fermentative acid production to rapid methane production.

## Overview of the landfill microbial community

1881 metagenome-assembled genomes (MAGs) were recovered from the landfill samples. The total number of MAGs recovered from older landfill cells was lower compared to newer landfill cells: 93 MAGs were recovered from cell A, 188 MAGs from cell B, 134 from cell C, 220 MAGs from cell D1, 269 from cell D2, 210 from cell E, 294 from cell F1 and 239 from cell F2 (Fig. 2).

MAGs were taxonomically classified to 62 phyla and 325 families (full taxonomy and annotation information for all MAGs is available in Supplementary Data 1 on the Open Science Framework (OSF) at

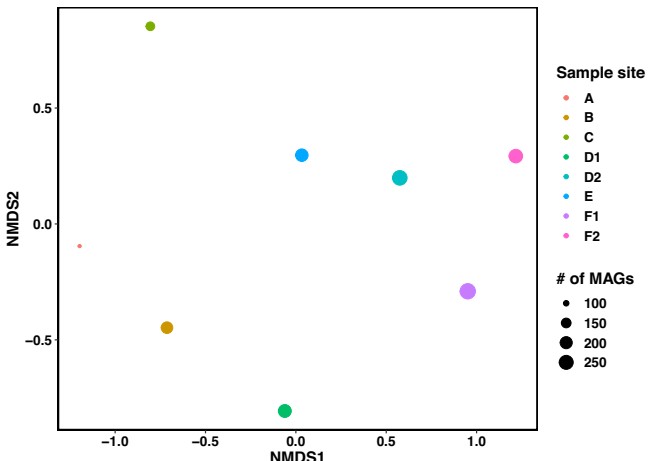

**Fig. 2 | Non-metric multidimensional scaling (NMDS) based on relative abundance summed at the family level for 1,647 metagenome-assembled genomes (MAGs) recovered from landfill leachate samples.** The Bray-Curtis dissimilarity index was used to generate a distance matrix required for the NMDS analysis. The default command 'metaMDS' from the 'vegan' package was used to run 20 iterations of the NMDS ordination, which provided a stress value of 0.0487. Landfill sampling sites have been colour-coded, and the size of each point has been scaled to the total number of MAGs recovered from each sample to indicate richness. Source Data are provided as a Source Data file.

https://doi.org/10.17605/OSF.IO/6X5ZC[37]). Beta diversity analyses based on relative abundance summed at the family level showed that community composition was more similar in newer landfill cells (i.e., D2, E, F1, and F2) compared to older landfill cells (i.e., cells A, B, and C) (Fig. 2). Although location D1 began operating only two years before location D2, the community composition of D1 was more similar to cell B, which began receiving waste 11 years earlier (Fig. 2). These differences in community composition for cell D are in line with the discrepancies in the composition of flare gas and the availability of organic carbon noted in the previous section. The community composition in cells A and B was more similar compared to cell C, which aligns with geochemical data indicating these parts of the landfill are in different lifecycle phases (Fig. 2). Overall, these observations suggest that newer landfill cells that have experienced methane production more recently support more diverse microbial populations with similar composition compared to older cells where methane production has declined.

Landfill cells D1, D2, E, F1, and F2 harboured more distinct phyla compared to cells A, B, and C, in line with the total number of MAGs recovered from each site (Fig S3). Microbial communities in cells A, B, and C harboured populations classified to the phylum *Proteobacteria* that ranged in abundance from 34 to 46%. In contrast, *Proteobacteria* were present at <5% in cells D1, D2, E, F1, and F2 (Fig S3). Members of the *Patescibacteria* ranged in abundance from 6 to 35% in cells B, D1, D2, and E, although no clear trend was observed for this lineage with respect to the age of each landfill cell (Fig S3). Members of the *Bacteroidota* displayed high relative abundance ranging from 15 to 39% in landfill cells producing methane or thought to be transitioning to a state of rapid methane production (i.e., D2, E, F1, and F2) (Fig S3). Members of the phylum *Campylobacterota* dominated landfill cell C with a relative abundance of 48% and occurred at the relative abundance of 28 and 22% in cells A and E, respectively (Fig S3). The relative abundance of the phyla *Firmicutes*_A, *Halobacterota* (renamed to *Halobacteriota* in GTDB r95), *Cloacimonadota*, and *Firmicutes* fluctuated with landfill cell age, ranging from 10 to 20% in newer landfill cells (i.e., D2, E, F1, and F2) (Fig S3). Most other phyla accounted for a minor portion of the community and occurred at low relative abundances, <1%, across the landfill (Fig S3).

Family-level data was interpreted by focussing on the most abundant families occurring across the landfill cells in chronological order based on age. Members of the *Gallionellaceae* ranged in abundance from 10 to 26% in older landfill cells (i.e., A, B, and C) but decreased to <5% in newer landfill cells (i.e., D1, D2, E, F1, and F2) (Fig S4). The abundance of *Gallionellaceae* aligns with oxygen intruding in older parts of the landfill given that representative genera such as *Gallionella* detected in the landfill contain exclusively microaerophilic iron oxidizers capable of autotrophic growth[38] well-suited to such habitats (Supplementary Data 1 on the Open Science Framework (OSF) at https://doi.org/10.17605/OSF.IO/6X5ZC[37]). Recent work has also shown that unknown species in the *Gallionellaceae* can support iron oxidation coupled to autotrophy and nitrate reduction[39], which aligns with the >90% completion of the Calvin cycle and the potential for denitrification observed in *Gallionellaceae* MAGs from the landfill (Supplementary Data 1 on the Open Science Framework (OSF) at https://doi.org/10.17605/OSF.IO/6X5ZC[37]). Members of the family *Sulfurimonadaceae* occurred at similar relative abundance to the *Gallionellaceae* (i.e., 26%) in cell A but were at much lower abundance in other landfill cells (i.e., B, C, D1, D2, E, F, and F1) (Fig S4). Members of the *Sulfurimonadaceae* harbour versatile redox metabolisms that can be coupled to autotrophy[40, 41], which would be well-suited to landfill cell A but also allow them to persist over redox gradients encountered in other landfill cells. These observations align with the capacity for thiosulphate oxidation and nitrogen reduction, and >70% completion of the Calvin cycle in *Sulfurimonadaceae* MAGs from the landfill (Supplementary Data 1 on the Open Science Framework (OSF) at https://doi.org/10.17605/OSF.IO/6X5ZC[37]).

The family *Arcobacteraceae* dominated the community at landfill cell C with a relative abundance of 44% (Fig S4). This dominance can be attributed to two populations of the genus *Aliarcobacter*, which displayed the highest genomic coverage for all MAGs analyzed in this data set (i.e., MAGs STC_123 and STC_124 had coverage values of 527.30 and 857.20, respectively) (Supplementary Data 1 on the Open Science Framework (OSF) at https://doi.org/10.17605/OSF.IO/6X5ZC[37]). Members of the *Aliarcobacter* genus (previously referred to as *Arcobacter*) have been detected in brackish waters, sewage, and food products (summarized in[42]) and have also been identified as abundant members of landfill microbial communities[8,14,17]. Members of this genus have been best-studied in the context of enteric and zoonotic pathogenesis[42,43], suggesting they could be introduced to landfills through food, animal, or human waste. Members of the *Aliarcobacter* can tolerate a wide range of changes in physical conditions and display the capacity for aerobic and microaerophilic growth (reviewed in[44]), which could make them well-suited to heterogeneous landfill environments. MAGs from the *Arcobacteraceae* in the landfill encoded the capacity for dissimilatory nitrate reduction, which may offer them an advantage in anoxic landfill habitats (Supplementary Data 1 on the Open Science Framework (OSF) at https://doi.org/10.17605/OSF.IO/6X5ZC[37]), however further investigation is required to identify which adaptations allow this family to dominate in landfill cell C.

Members of the *Dysgonomonadaceae* and *Cloacimonadaceae* displayed comparable relative abundance to the *Gallionellaceae* and *Sulfurimonadaceae* (i.e., 12 to 20%) but only in newer landfill cells in states of high methane production (i.e., E, F1, and F2) (Fig S4). Members from both families have repeatedly been detected in landfills[17,35,45] and could potentially contribute substrates to support methanogenesis. Previous work on members of the *Dysgonomonadaceae* in anaerobic digestion settings has shown that members of this family can hydrolyze recalcitrant organic substrates to produce acetate, which can be further metabolized to produce carbon dioxide and hydrogen during anaerobic fermentative growth[46–48]. Recent genomic surveys of the phylum *Cloacimonadota*, in which *Cloacimonadaceae* is the sole named family, have shown there are distinct clades adapted to landfills that can potentially support methane production through acetogenic metabolism[45]. These observations align with the metabolic potential observed in MAGs classified to the *Dysgonomonadaceae* and *Cloacimonadaceae* wherein most MAGs included genes coding for the phosphotransacetylase and acetate kinase that could be used to produce acetate (Supplementary Data 1 on the Open Science Framework (OSF) at https://doi.org/10.17605/OSF.IO/6X5ZC[37]). Although most MAGs from the *Dysgonomonadaceae* were largely classified to the genera *Proteiniphilum* and *Fermentimonas*, MAGs from the *Cloacimonadaceae* could not be classified to the genus level (Supplementary Data 1 on the Open Science Framework (OSF) at https://doi.org/10.17605/OSF.IO/6X5ZC[37]). This observation suggests there is still considerable diversity to uncover in this lineage with important roles in carbon cycling in landfills.

## Methanogen community structure

From the initial 1,881 MAGs obtained from landfill metagenomes (Methods and Table S2 for metagenome statistics), 74 MAGs were identified as putative methanogens coming from leachate samples. The putative methanogen MAGs were taxonomically classified into 3 phyla the following curation: *Halobacterota* (renamed to *Halobacteriota* in newer versions of GTDB), *Thermoplasmatota*, and *Euryarchaeota* (Fig. 3). These phyla spanned 10 families (in alphabetical order): *Methanobacteriaceae, Methanocorpusculaceae, Methanocullaceae, Methanofollaceae, Methanomethylophilaceae, Methanomicrobiaceae, Methanoregulaceae Methanosarcinaceae, Methanospirillaceae*, and *Methanotrichaceae* (Figs. 3, S5). In this instance, we did not classify MAGs from the family *Methanoperedenaceae* as methanogens, given that they are known anaerobic methane oxidizing Archaea, and are instead discussed in the methanotrophy section (Figs. 3, S5). Two MAGs could not be classified to the family level: STE_86, classified to the order *Methanobacteriales*, and STF1_149, the only MAG classified to the order *Methanofastidiosales* (Figs. 3, S5, and Supplementary Data 1 on the Open Science Framework (OSF) at https://doi.org/10.17605/OSF.IO/6X5ZC[37]).

Methanogenic families occurred at low relative abundance across the landfill. Members of the *Methanocorpusculaceae* and *Methanocullaceae* had the highest relative abundance values of ~6% in newer parts of the landfill (i.e., D2, F1, and F2) whereas most other methanogen families occurred at relative abundance ~2% or lower across the landfill (Figs. 3, S5). Methanogen MAGs tended to have higher relative abundance compared to the mean and median values for the whole community in cells D2, F1, and F2 rather than cells A, B, and C, which aligns with the higher methane production observed for newer landfill cells (Figs. 1, 3, and S5). Higher numbers of MAGs from different methanogenic families were also recovered from cells D2, F1, and F2, compared to cells A, B, and C suggesting these parts of the landfill also support more diverse methanogenic communities (Figs. 3, S5). Cells D1 and E displayed intermediate trends with respect to relative abundance and diversity that did not align with geochemical observations, particularly for E, which shows consistently high methane production. These two cases are discussed in detail in the subsequent sections where we examine the taxa and metabolic pathways that could be contributing to methane gradients in the landfill.

Landfill cells A, B, and C contained two, three, and seven putative methanogen MAGs, respectively. These MAGs were classified to the families *Methanocorpusculaceae, Methanocullaceae, Methanomethylophilaceae, Methanoregulaceae*, and *Methanotrichaceae* (Fig. 3 and Fig. S5). The relative abundance of putative methanogens was at least an order of magnitude lower than the most abundant MAGs from each site, which included members of the *Sulfurimonadaceae* (i.e., STA_16 at 15.81%), the *Gallionellaceae* (i.e. STB_49 at 10.84%), and *Arcobacteraceae* (i.e., STC_123 at 27.17%) as noted previously (Fig. 3, Fig S4, and Supplementary Data 1 on the Open Science Framework (OSF) at https://doi.org/10.17605/OSF.IO/6X5ZC[37]).

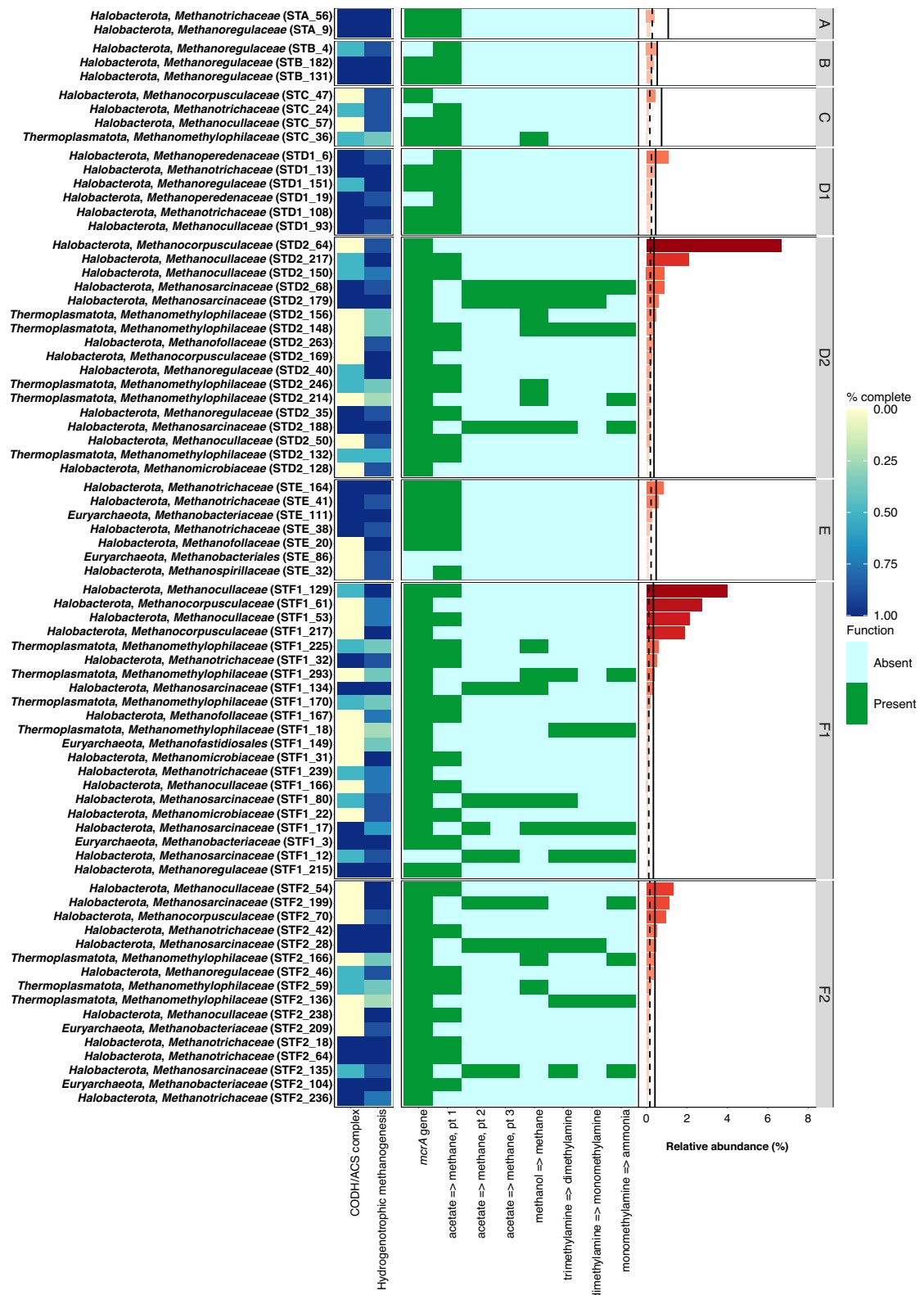

**Fig. 3 | Metabolic heatmap and relative abundance data for putative methanogenic metagenome-assembled genomes (MAGs).** The abbreviation CODH/ACS complex denotes the completion of the carbon monoxide dehydrogenase/acetyl-CoA synthase pathway. Relative abundance values have been scaled to the most abundant MAG classified to a methane cycling guild. Solid black vertical lines denote the mean relative abundance calculated for MAGs at the whole-community level and dashed black lines denote the median relative abundance calculated for MAGs at the whole-community level. Source Data are provided as a Source Data file.

According to classification rules used in this study (see Methods), all putative methanogen MAGs in landfill cells A and B can be classified as acetoclastic methanogens based on the presence of the acetyl-CoA synthetase (i.e., noted as acetate => methane pt.1 by DRAM), high completion (i.e., >50%) of the carbon monoxide dehydrogenase/acetyl-CoA synthase (CODH/ACS) complex supporting acetyl-CoA dismutation, and high completion (i.e., >75%) of the hydrogenotrophic methanogenesis pathway (Fig. 3). The functional potential for methanogenesis from cell C shows a mix of strictly hydrogenotrophic, acetoclastic, and methylotrophic methanogens (Fig. 3, see Methods).

When considering the metabolic analyses alongside our predictions based on geochemistry, the restricted pathways that can support methanogenesis align with observations of these habitats having limited substrates available that can support methanogenesis. The abundance of microaerophilic autotrophic guilds further supports that oxygen infiltration may be inhibiting methanogenesis in these parts of the landfill, which likely contributes to the lower diversity and abundance of methanogens observed.

Looking at the microbial community from locations D1 and D2, we see contrasting trends in the putative methanogenic community. These trends emulate what was observed at the whole-community scale and suggest cell D is not experiencing homogenous geochemistry that is impacting methanogenesis (Figs. 2, 3, and Fig S5). The putative methanogenic community from D1 is comprised of six MAGs that include many families detected in cells A, B, and C, including the *Methanotrichaceae, Methanoregulaceae*, and *Methanocullaceae* (Fig. 3 and Fig S5). An additional two MAGs from the *Methanoperedenaceae* were also recovered from D1 based on the detection of near complete hydrogenotrophic methanogenesis pathways (i.e., STD1_6 and STD1_19) (Fig. 3 and Supplementary Data 1 on the Open Science Framework (OSF) at https://doi.org/10.17605/OSF.IO/6X5ZC[37]). The detection of the *Methanoperedenaceae* is noteworthy as they are the only named family of anaerobic methane-oxidizing Archaea (ANME) and these MAGs are discussed within the examination of methanotrophy, below. The relative abundance of putative methanogens was an order of magnitude lower compared to the most abundant MAG STD1_23, which had an abundance of 4.19% and was classified to the unnamed family UBA6257 in the phylum *Patescibacteria* (Supplementary Data 1 on the Open Science Framework (OSF) at https://doi.org/10.17605/OSF.IO/6X5ZC[37]).

In contrast to location D1, 17 putative methanogen MAGs were recovered from location D2. There was considerable overlap between the families detected in D2 and cells A, B, C, and location D1, with four additional families identified at location D2 including the *Methanobacteriaceae, Methanofollaceae, Methanomicrobiaceae*, and *Methanosarcinaceae* (Figs. 3, S5). D2 was unique among landfill cells because MAG STD2_64 from the family *Methanocorpusculaceae* displayed the highest relative abundance of 6.66% among all MAGs at location D2 (Figs. 3, S4, S5). Additional MAGs from the *Methanocullaceae* (e.g., STD2_217 and STD2_150) and *Methanosarcinaceae* (e.g., STD2_68 and STD2_179) had lower relative abundance values ranging from 0.5 to 2% but were still more abundant than most other microbial taxa identified at this location (Figs. 3, S5).

The differing trends observed in cells D1 and D2 extend to the methanogenesis pathways predicted at both locations. The capacity for methanogenesis at D1 resembles that of cells A and B, with MAGs from the *Methanotrichaceae, Methanoregulaceae*, and *Methanoculla-ceae* all classified as acetoclastic methanogens based on the presence of genes coding for the acetyl-CoA synthetase, >50% complete CODH/ACS complexes, and >75% complete hydrogenotrophic methanogenesis pathways (Fig. 3). More substrates can potentially support methane production at D2. MAGs from the *Methanocorpusculaceae, Methanofollaceae, and Methanomicrobiaceae* were classified as strictly hydrogenotrophic methanogens (Fig. 3). MAGs from the *Methanocullaceae* and *Methanoregulaceae* were classified as acetoclastic

methanogens and MAGs from the *Methanomethylophilaceae* were classified as methylotrophic methanogens. The broadest capacity for methanogenesis was observed in MAGs from the family *Methanosarcinaceae* (i.e., STD2_68, STD2_179, and STD2_188), which displayed the potential for hydrogenotrophic, acetoclastic, and methylotrophic methanogenesis (Fig. 3).

Despite initially classifying both locations in cell D as being in phase 4 of slow methanogenesis, our microbiological observations suggest biogeochemical succession is occurring in a more segregated fashion. Although location D1 houses waste that is only 2 years older than location D2, the whole community structure and methanogenic capacity observed at D1 are more in line with landfill cells that are 10 years older. This discrepancy may be occurring due to cell D's connection to the C/D valley location where oxygen intrusion is suspected (Fig. 1B). Alternatively, the location of the gas flaring vent associated with location D2 may not capture a representative signal of the prevailing geochemistry of this habitat.

Methanogens were the most abundant population in D2 and demonstrated broader methanogenic metabolic capabilities suggesting substrates that can support methanogenesis may be more widely available at D2 compared to D1. Concentrations of inorganic carbon were higher at D2 compared to D1 in the timeframe surrounding our sampling expedition (i.e., ~2500 to 3500 mg L$^{-1}$ vs ~1000 mg L$^{-1}$), which may be contributing to the abundance of hydrogenotrophic methanogens at D2 (Fig. 3 and Fig S2). As previously noted, acetate concentrations were also an order of magnitude higher at D2 vs. D1 prior to sampling, which could also favour acetoclastic methanogens. These observations suggest that D2 is transitioning from phase 3 to phase 4 and frame D2 as a potential methane production hotspot whose contributions may be masked when examining bulk gas flaring data.

Landfill cell E was originally classified as being in phase 3 of rapid methanogenesis. The low abundance and diversity of putative methanogens in cell E is at odds with the original classification based on geochemistry. Only seven putative methanogenic MAGs were recovered from landfill cell E, from the families *Methanotrichaceae*, also observed in cells A, C, and location D1; the *Methanofollaceae* and *Methanobacteriaceae* also observed in D2; and the *Methanospirillaceae*, which was not detected anywhere else in the landfill (Figs. 3, S5). One MAG could only be classified to the *Methanobacteriales* order (Figs. 3, S5, and Supplementary Data 1 on the Open Science Framework (OSF) at https://doi.org/10.17605/OSF.IO/6X5ZC[37]). The relative abundance of putative methanogens was an order of magnitude lower compared to the most abundant MAG STE_65, which had an abundance of 8.77% and was classified to the family *Sulfurovoraceae* (Figs. 3, S4, S5, and Supplementary Data 1 on the Open Science Framework (OSF) at https://doi.org/10.17605/OSF.IO/6X5ZC[37]).

The methanogenic community in cell E resembles those associated with waste that is 11 to 19 years older (i.e., cells A, B, and C) despite landfill cell E displaying a similar whole community composition to D2, F1, and F2 (Figs. 2, 3, S5). The methanogenic capabilities potentially supported in landfill cell E were restricted to acetoclastic and hydrogenotrophic methanogenesis despite lower methane-producing areas of the landfill, such as D2, showing broader methanogenic pathways (Figs. 1, 3). The occurrence of acetoclastic and hydrogenotrophic methanogens in cell E aligns with our initial prediction that a limited number of substrates capable of supporting methane production would be available at this stage in the landfill's lifecycle; however, the observed low abundance and diversity of methanogens was unexpected given what was observed at location D2.

From a microbiological perspective, these observations suggest cell E has reached the end of phase 3 and will soon enter phase 4 of slow methanogenesis. The revised classification of cell E is difficult to reconcile with the consistently high methane production associated with this part of the landfill. This discrepancy between the methanogenic community structure and the whole community structure in cell

E is notable because it suggests that methanogenic guilds are sensitive to a changing variable in landfill cell E.

Potential explanations are threefold: First, the abundance of putative methanogens may be decoupled from their metabolic activity. However, we would expect increased rates of methanogenesis to translate into more biomass and/or genomes given that methanogenesis is energy-conserving. Second, the sharp declines in inorganic carbon that occurred just prior to sampling may have caused a decline in acetoclastic and hydrogenotrophic methanogens using inorganic carbon for methane production (Fig S2). Finally, our characterization of the methanogen community from leachate at a more local scale may not align with gas data obtained for the entire landfill cell. Given that cell E is the only cell that experienced extreme fluctuations in substrates that could support methane production at the time of sampling, we're inclined to attribute observed disparities to the availability of bicarbonate, though this mechanism needs to be formally tested in the future.

Landfill cell F was originally classified as transitioning from phase 2 of anaerobic acid production to phase 3 of rapid methanogenesis. In contrast to the samples obtained from both locations in cell D, the abundance, diversity, and community structure of putative methanogens were more consistent between both locations sampled from cell F despite their two-year age difference. 20 putative methanogen MAGs were recovered from location F1 and 17 putative methanogen MAGs were recovered from location F2 (Fig. 3). The families detected at landfill cell F (i.e., *Methanobacteriaceae*, *Methanocorpusculaceae*, *Methanocullaceae*, *Methanofollaceae*, *Methanomethylophilaceae*, *Methanomicrobiaceae*, *Methanoregulaceae*, *Methanosarcinaceae*, and *Methanotrichaceae*) were all detected in the landfill cells discussed previously (Figs. 3, S5). There was considerable overlap in the families detected in F1 and F2 aside from location F1 harbouring MAGs from the *Methanofollaceae* and *Methanomicrobiaceae*, families not detected at F2 (Figs. 3, S5).

Putative methanogen MAGs from location F1 ranged in relative abundance from 0.05 to 3.97% whereas those from F2 displayed a slightly lower range of 0.08 to 1.33% (Figs. 3, S5). The abundance of putative methanogens at F1 was lower than the most abundant MAG at location F1, MAG STD1_175, which was classified to the family *Cloacimonadaceae* and had a relative abundance of 7.79% (Figs. S3, S4, S5, and Supplementary Data 1 on the Open Science Framework (OSF) at https://doi.org/10.17605/OSF.IO/6X5ZC[37]). The abundance of putative methanogens at F2 was almost an order of magnitude lower compared to the most abundant MAG, MAG STF1_112, which was also classified to the family *Cloacimonadaceae* and had a relative abundance of 9.42% (Figs. S3, S4, S5, and Supplementary Data 1 on the Open Science Framework (OSF) at https://doi.org/10.17605/OSF.IO/6X5ZC[37]).

The putative methanogenic communities from locations F1 and F2 displayed similar pathways that could support methanogenesis, with hydrogenotrophic and acetoclastic pathways represented in multiple families. All MAGs from the family *Methanosarcinaceae* could be classified as broad substrate methanogens possessing near complete pathways for acetoclastic and hydrogenotrophic methanogenesis alongside genes required to convert methyl-bearing substrates to methane (i.e., STF1_134, STF1_80, STF1_17, STF1_12, STF2_199, STF2_28, STF2_135) (Fig. 3). The predicted capacity for multiple methanogenic pathways in members of the *Methanosarcinaceae* in cells F and D suggest members of this lineage are generalists capable of accessing a variety of substrates to support methane production over the course of a landfill's lifecycle. The absence of the *Methanosarcinaceae* in cells A, B, and C also suggests that generalists from this family are outcompeted by more specialized methanogens as MSW ages (Figs. 3, S5). Alternatively, this absence could be attributed to different acetate affinities. Members from families such as the *Methanotrichaceae* have a higher affinity for acetate and would have a physiological advantage in ageing landfills with μM amounts[49], whereas the members of the

*Methanosarcinaceae* have a growth advantage and outcompete other acetoclastic methanogens at the mM levels more likely to be available in newer landfills.

The sole MAG classified to the order *Methanofastidiosales* (i.e., STF1_149) carried a homolog for the *mtsA* gene coding a methyltransferase specific to methylthiol-bearing compounds (see Supplementary Note 1). Members of this order are inferred to carry out methanogenesis in a fastidious manner via the reduction of methylthiol and we predict the same metabolism here[50].

We note MAGs from the *Methanocullaceae* family were mixed as to their classification as hydrogenotrophic or acetoclastic methanogens across multiple landfill cells based on the rules we used to classify putative methanogenic metabolisms. Physiological studies on isolates from the *Methanoculleus* the only genus from the *Methanocullaceae* recovered from the landfill (Supplementary Data 1 on the Open Science Framework (OSF) at https://doi.org/10.17605/OSF.IO/6X5ZC[37]), have shown that acetate is a required carbon source that does not serve as a substrate for methane production under the conditions tested[51–57]. Members from the *Methanobacteriaceae* were subject to a similarly mixed classification in our study despite previous work showing that acetate is not a methanogenic substrate (compiled in[58]). These discrepancies highlight the limits of purely metagenomic approaches, which cannot distinguish between whether the same metabolic machinery contributes to carbon assimilation and/or energy conservation via methanogenesis[59]. Similar limitations must be considered for methanogens that carry genes for hydrogenotrophic pathways to support the disproportionation of methyl-bearing substrates used for methane production[60,61]. Such limitations could be resolved by combining metagenomic approaches with metatranscriptomic and isotopic approaches with labelled substrates in future studies.

The abundance and diversity of methanogens in cell F align with our original classification of this landfill cell as transitioning from phase 2 of anaerobic acid production to phase 3 of rapid methanogenesis. Despite the peaks in organic acids being considerably lower in cell F compared to older landfill cells, the presence of varied substrates may support rapid methanogenesis due to limited competition for carbon substrates among methanogens (Fig S2). We posit that generalists and specialists alike have ample resources to support methanogenesis and contribute to high rates of methane production in cell F. This position is supported by the higher relative abundance of taxa that can potentially supply acetate to methanogens in these parts of the landfill, although a full characterization of potential syntrophic relationships is outside the scope of this study (see Supplementary Note 2 and Fig S6).

## Methanogen families occurring in landfills

To place the methanogenic taxonomic diversity found at our study site in context, we generated a compilation of presence/absence data for methanogenic taxa identified in landfills in the past 20 years (Fig. 4). Our study site harbours some of the most diverse communities of methanogens reported to date (i.e., 11 families detected). Only a 16S rRNA amplicon sequencing survey conducted across six distinct landfills in China showed higher diversity (i.e., 14 families detected, see Fig. 4), which displayed considerable overlap in the families observed at our site[20]. The remaining studies compiled, many of which also analyzed more than one landfill, detected lower taxonomic diversity regardless of the method employed (Fig. 4).

Members from the *Methanosarcinaceae* are the most frequently detected in landfills, occurring in 17 of the 21 studies included (Fig. 4). Members of the *Methanosarcinaceae* are widespread in terrestrial, aquatic, and animal-associated habitats due to their ability to use multiple substrates to support methanogenesis[62,63]. This generalist strategy likely contributes to their occurrence in landfills covering a range of ages and geographic locations, and frames members of the *Methanosarcinaceae* as key players in the landfill methane cycle (see Supplementary Data 2 on the Open Science Framework (OSF) at

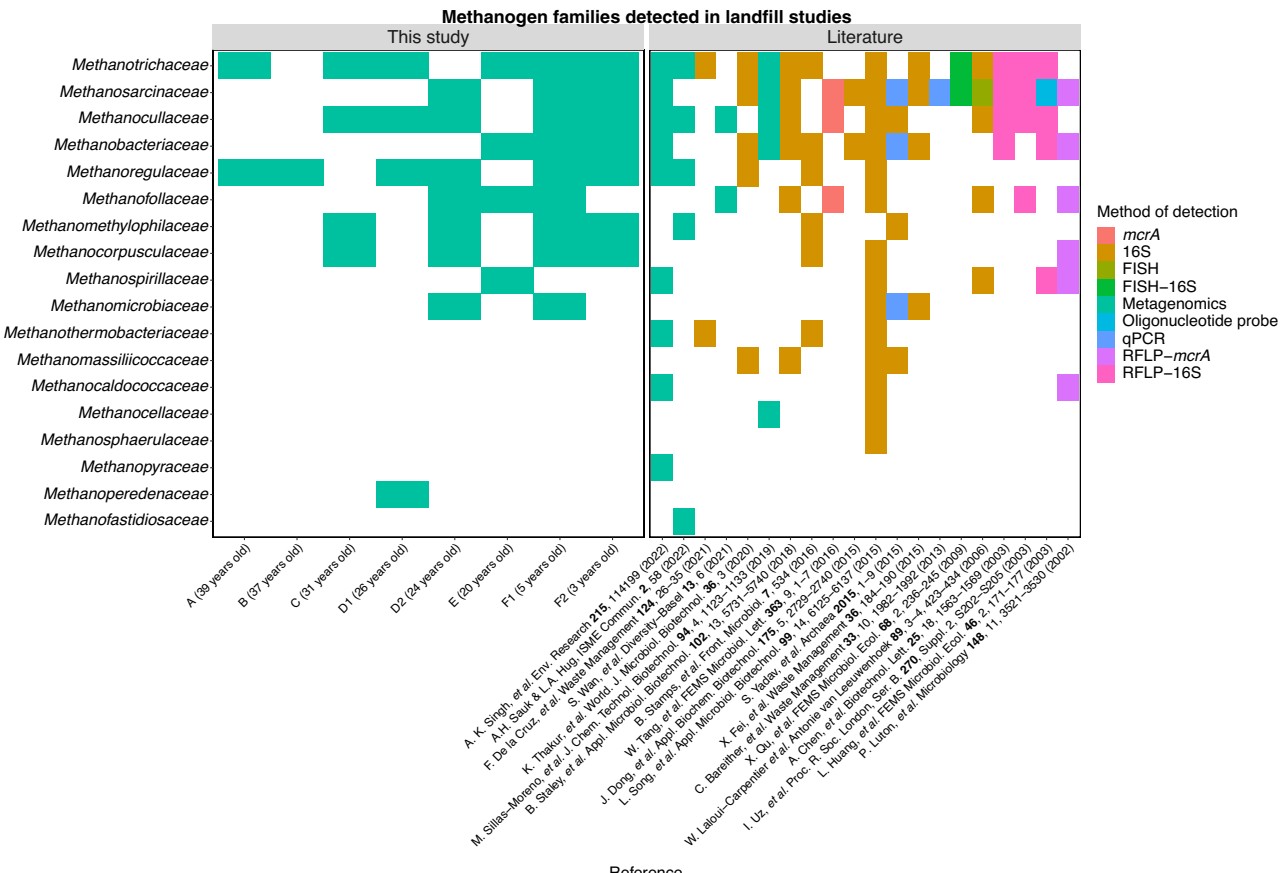

**Fig. 4 | Methanogen and *mcr*-bearing families detected in this study and previously published work using different analytical techniques.** Where possible, all taxonomic labels have been updated to reflect the Genome Taxonomy Database (GTDB) r89 taxonomy used in our study. Abbreviations denote the following: 16S (shorthand for 16S rRNA amplicon sequencing), FISH (fluorescent in situ hybridization), RFLP (restriction fragment length polymorphism), and qPCR (quantitative polymerase chain reaction). Abbreviated references have been provided and the raw input data used to generate this figure can be found in Supplementary Data 4 on the Open Science Framework (OSF) at https://doi.org/10.17605/OSF.IO/6X5ZC[37]. Data obtained from (in chronological order):[8,15,16,18–20,22,23,35,36,68,117–122]. Source Data are provided as a Source Data file.

https://doi.org/10.17605/OSF.IO/6X5ZC[37] for full accounting of methanogenic lineages in landfills).

After the *Methanosarcinaceae*, the *Methanotrichaceae* (previously referred to as the *Methanosaetaceae* in the literature) (15/21 studies), *Methanocullaceae* (13/21 studies), and *Methanobacteriaceae* (13/21 studies) families were the next most frequently detected in landfills (Fig. 4). Members of the *Methanotrichaceae* are considered acetoclastic methanogens in line with our metabolic modeling[62] and we have already noted the potential limitations of members of the *Methanocullaceae* and *Methanobacteriaceae* as acetoclastic methanogens in the absence of physiological evidence. Regardless of which substrates support methane production, these families were repeatedly detected in newer waste but also waste that was over 20 years old, including in our data, suggesting they are important contributors to long-term methane production in landfills (Supplementary Data 3 on the Open Science Framework (OSF) at https://doi.org/10.17605/OSF.IO/6X5ZC[37]).

The persistence of families such as the *Methanotrichaceae* and *Methanocullaceae* in older waste could be attributable to their increased tolerance to oxidative stress[64], an important adaptation to heterogeneous landfill habitats subject to large fluctuations in redox potential. Indeed, many of the methanogen families detected in landfills fall within orders forming a distinct clade of methanogens whose genomes are enriched in oxidative stress tolerance genes (e.g., *Methanocorpusculaceae*, *Methanofollaceae*, *Methanomicrobiaceae*, *Methanoregulaceae*, *Methanosarcinaceae*, *Methanosphaerulaceae*, *Methanospirillaceae*)[64] (Fig. 4).

Families containing more metabolically restricted methanogens were less frequently detected in landfills. Members of the *Methanofollaceae* and *Methanocorpusculaceae*, considered to be strictly hydrogenotrophic methanogens, were detected in 8 and 4 of 21 studies, respectively[65–67] (Fig. 4). Methylotrophic methanogens from the family *Methanomethylophilaceae* were detected in 4 of 21 studies and methanogens from the family *Methanofastidiosaceae* from the order *Methanofastidiosales*, which are inferred to use methylthiols in methane production, were only detected in one study from our group, suggesting landfill habitats are generally not conducive to supporting this specific methanogenic lifestyle (Fig. 4)[35,50,68].

These observations do not preclude metabolically restricted methanogens from being important contributors to methane production in landfills. Strictly hydrogenotrophic methanogens were some of the most abundant MAGs recovered from location D2 and likely play an important role in producing methane as organic substrates are depleted. Likewise, multiple MAGs of methylotrophic methanogens from the *Methanomethylophilaceae* family were detected in cell F suggesting they are active contributors to methane production earlier in the landfill lifecycle when labile organic substrates are more likely to be available. We note that our categorization system did not distinguish between the $H_2$-dependent methylotrophic methanogenesis pathway that oxidizes methylated compounds to methane and methylotrophic methanogens that rely on the disproportionation of methylated compounds to form methane[63]. The reported capacity for $H_2$-dependent methylotrophic methanogenesis

within the order *Methanomassiliicoccales*[69–71], which contains the family *Methanomethylophilaceae*, highlights the importance of considering the dual contributions of methylotrophic methanogens to hydrogen and methane cycling in landfills. The contributions of methylthiol-using methanogens to the methane cycle remain poorly characterized. The detection of members of the *Methanofastidiosales* in two different landfill studies suggests they also contribute to methane production in MSW.

## Methanotrophic community structure

Our initial survey of MAGs containing the *pmoA* and/or *mmoX* genes identified 31 MAGs as putative aerobic methanotrophs, with the addition of the 2 ANME MAGs from D1 as putative anaerobic methanotrophs. Within the aerobic methanotrophic MAGs, 15 encoded only *pmoA*, 5 encoded only *mmoX*, and 11 carried genes for both pMMO and sMMO complexes (Fig. 5). Almost all putative methanotroph MAGs were found in parts of the landfill where oxygen was detected or suspected to be intruding (i.e., cells A, B, C, and location D1) (Figs. 1B, 5).

Putative methanotroph MAGs spanned all three phyla with known methanotrophic representatives: the *Proteobacteria*, *Verrucomicrobiota*, and *Methylomirabilota* (Figs. 5, S5)[72, 73]. Of these, 25 MAGs belonged to families associated with aerobic or intra-aerobic methanotrophy, including the *Methylomonadaceae* (20), *Methylococcaceae* (2), *Methylacidiphilaceae* (2), and *Methylomirabilaceae* (1) (Figs. 5, S5). Six MAGs were recovered from lineages whose capacity for methanotrophy remains poorly characterized or untested. These MAGs were taxonomically classified within the *Proteobacteria* families *Acetobacteraceae* (i.e., STB_66 and STC_13) and *Nevskiaceae* (STE_114), in addition to the phyla *Elusimicrobiota* (STA_59), *Actinobacteriota* (STB_95), and *Chloroflexota* (STD1_5) (Figs. 5, S5).

Methanotrophic families occurred at low relative abundance compared to the rest of the community and a slightly lower relative abundance compared to methanogens (Fig. S5). Members of the *Methylomonadaceae* had relative abundance values between ~2-4% in cells B and D1 but were generally at ~1% or lower in the remaining landfill cells where methanotrophs were detected (i.e., A, C, and E) (Figs. 5, S5). Most other methanotroph families occurred at relative abundances between 0.1 to 1% (Fig. 5, S5). The low abundance of methanotrophs and methanogens in all locations except D2 suggests that methane cycling guilds are not dominant members in many of the landfill cells sampled. Despite the low abundance of methanotrophs, we sought to better understand the physiological adaptations they exhibit in landfill habitats where oxygen is limited, given the potential for limiting methane emissions through their activities.

In landfill cell A, four putative methanotroph MAGs were detected. Three MAGs belonged to the family *Methylomonadaceae* (i.e., STA_49, STA_13, and STA_76) and one belonged to the unnamed family UBA9628 from the phylum *Elusimicrobiota* (STA_59) (Figs. 5, S5). Methanotroph MAGs displayed low abundances (<1%) that were comparable to methanogens, suggesting methane cycling is balanced but not a dominant process in this landfill cell (Fig S5).

Putative methanotroph MAGs were more abundant and diverse in landfill cell B compared to cell A, ranging in abundance from 0.12 to 2.36% with 12 MAGs recovered (Figs. 5, S5). Ten of these MAGs were classified to families with known methanotrophic activities [i.e., *Methylomonadaceae* (8), *Methylococcaceae* (1), and *Methylacidiphilaceae* (1)] wherein the family *Methylomonadaceae* was the most abundant (Figs. 5, S5)[73]. The remaining two MAGs were classified to the *Acetobacteraceae* from the phylum *Proteobacteria* (STB_66) and the family *Mycobacteriaceae* from the phylum *Actinobacteriota* (STB_95) (Figs. 5, S5). MAG STB_122 from the *Methylomonadaceae* was the most abundant putative methanotroph recovered with an abundance of 2.36%, which suggests they may be important contributors to methanotrophy in landfill cell B (Fig. 5).

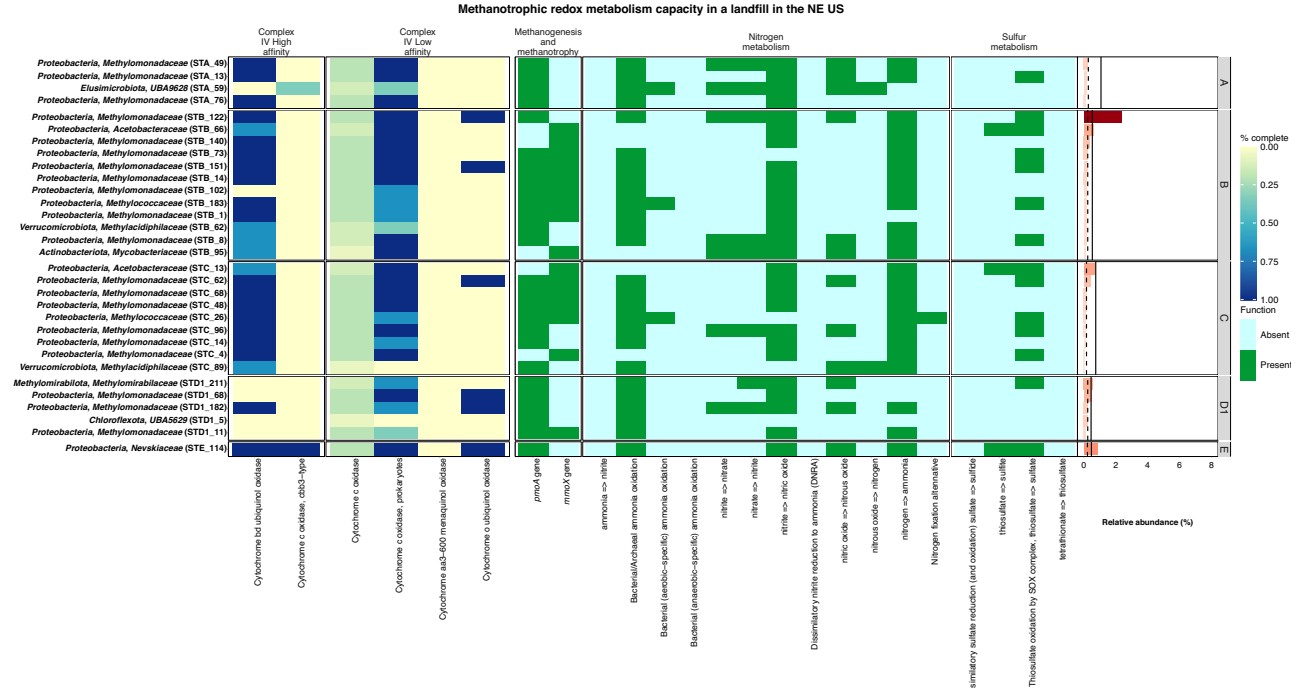

**Fig. 5 | Metabolic heatmap and relative abundance data for putative methanotrophic metagenome-assembled genomes (MAGs).** Shown here is the completion of high and low-affinity complex IV machinery involved in reducing oxygen alongside the presence/absence of genes involved in methane metabolism, nitrogen metabolism, and sulphur metabolism. No putative methanotroph MAGs were recovered from locations D2, F1, and F2, and as such no panels for those sampling locations are shown. Relative abundance values have been scaled to the most abundant MAG classified to a methane cycling guild. Solid black vertical lines denote the mean relative abundance calculated for MAGs at the whole-community level and dashed black lines denote the median relative abundance calculated for MAGs at the whole-community level. Source Data are provided as a Source Data file.

The putative methanotrophic community from cell C displayed a similar pattern to cell B with respect to diversity and low relative abundance values ranging from 0.072 to 0.67% (see Figs. 5, S5). Nine putative methanotrophic MAGs were recovered from cell C. Eight of these MAGs were classified to families with known methanotrophic activities [i.e., *Methylomonadaceae* (6), *Methylococcaceae* (1), and *Methylacidiphilaceae* (1)][73] and one MAG was classified to the family *Acetobacteraceae* (STC_13) (Figs. 5, S5). MAG STC_13 from the *Acetobacteraceae* had the highest relative abundance among putative methanotrophs at 0.67% but all methanotrophs had lower relative abundance compared to the mean for the whole community in cell C (Figs. 5, S5).

The detection of two *Methylacidiphilaceae* populations in cells B and C is of note because, outside of two sequencing studies that detected this family in a corroding sewer pipe[74] and landfill soils[36], the *Methylacidiphilaceae* have been largely characterized in acidic geothermal habitats (compiled in[75]). Phylogenomic analyses show that MAGs STB_62 and STC_89 are distinct from each other and clustered within the *Methylacidimicrobium* genus, which is considered to be a "mesophilic" clade within the *Methylacidiphilaceae*[75–78] (Fig S7). Metabolic reconstructions identified the presence of urease genes in the two landfill *Methylacidiphilaceae* MAGs that were absent in other *Methylacidiphilaceae* genomes (full *Methylacidiphilaceae* annotations provided in Supplementary Data 4 on the Open Science Framework (OSF) at https://doi.org/10.17605/OSF.IO/6X5ZC[37]). The capacity to hydrolyze urea may represent an adaptation to landfills where the hydrolysis of urea could fulfil the dual role of providing a nitrogen source and inorganic carbon to support biomass production through the Calvin-Benson-Bassham cycle, as seen in geothermal representatives of the *Methylacidiphilaceae*[76,79] (full methanotroph annotations provided in Supplementary Data 5 on the Open Science Framework (OSF) at https://doi.org/10.17605/OSF.IO/6X5ZC[37]). Our observations add evidence challenging the view that members of the *Methylacidiphilaceae* are exclusively acidophiles and further support that this family exists in temperate circumneutral habitats where methane is available.

Sampling location D1 displayed a lower abundance and diversity of putative methanotroph MAGs compared to cells B and C, with three MAGs classified to the *Methylomonadaceae*, one classified to the unnamed family UBA5629 in the phylum *Chloroflexota* (STD1_5), one classified to the family *Methylomirabilaceae* (STD1_211), and the two ANME *Methanoperedenaceae* MAGs (Figs. 3, 5). MAG STD1_6 from the *Methanoperedenaceae* was the most abundant putative methanotroph MAG with a relative abundance of ~1% exceeding the mean and median values reported for the whole community whereas most other methanotroph MAGs were below the mean and median relative abundance (Figs. 5, S5). No methanotrophic lineages were identified in location D2.

Finally, landfill cell E harboured a single putative methanotroph MAG from the family *Nevskiaceae* (STE_114) (Figs. 5, S5). This MAG occurred at a low abundance of 0.84% that was similar to the mean and median relative abundance for the whole community and was identical to the most abundant putative methanogenic MAG, STE_164 from the family *Methanotrichaceae* (Figs. 3, 5, S5). To the best of our knowledge, members of the *Nevskiaceae* family have yet to be tested for their ability to carry out methanotrophy through the pMMO pathway. No methanotrophic lineages were identified from cell F.

## Expanding the breadth of methanotrophic niches in landfills

The occurrence of the *Methylomirabilaceae*, the only family thought to support intra-aerobic methane oxidation, alongside the *Methanoperedenaceae* (Fig S5), suggests landfills (here location D1) are conducive to methanotrophic lifestyles that do not rely on exogenous oxygen. Although previous work has demonstrated the anaerobic oxidation of methane (AOM) in landfill-cover soil microcosms and anoxic landfill sites, the contributions of anaerobes to methane oxidation remain understudied compared to aerobes in cover soils[80,81]. We identified *Methanoperedenaceae* and *Methylomirabilaceae* populations that can potentially contribute to AOM in the landfill and metabolic adaptations in putative aerobic methanotrophs that may support methane oxidation via the pMMO and sMMO pathways coupled to anaerobic metabolism.

Phylogenomic analyses of the *Methanoperedenaceae* showed that the two MAGs recovered from the landfill were distinct from each other, clustering with unnamed *Methanoperedens* species sequenced from contaminated groundwater and bioreactor enrichments (Fig S8). MAG STD1_6, which had the highest relative abundance among all *mcr*-containing MAGs recovered from location D1 (Fig. 3), may represent a member of the *Methanoperedenaceae* with distinct adaptations to contaminated aquatic habitats that merit further investigation.

Previous work has shown that members of the *Methanoperedenaceae* can oxidize methane via reverse methanogenesis coupled with the reduction of metals, oxidized nitrogen, and oxidized sulphur with the help of sulphate-reducing partners[82–85]. The *Methanoperedenaceae* MAGs recovered from the landfill displayed differing redox metabolisms that could potentially be coupled to AOM. MAG STD1_6 lacked the potential to reduce oxidized forms of nitrogen and sulphur, like the unnamed species *Methanoperedens* sp. 902386115, which is curious given that *Methanoperedens* sp. 902386115 is more closely related to MAG STD1_19 (Fig S8 and Supplementary Data 5 on the Open Science Framework (OSF) at https://doi.org/10.17605/OSF.IO/6X5ZC[37]). MAG STD1_19 demonstrated the potential to reduce nitric oxide to nitrous oxide, a trait observed in 7 of the 13 *Methanoperedenaceae* genomes analyzed (Supplementary Data 5 on the Open Science Framework (OSF) at https://doi.org/10.17605/OSF.IO/6X5ZC[37]). MAG STD1_19 lacked the genes required for other steps in denitrification, suggesting it would require a metabolic handoff to acquire nitric oxide as a metabolic substrate to support AOM (Supplementary Data 5 on the Open Science Framework (OSF) at https://doi.org/10.17605/OSF.IO/6X5ZC[37]).

The *Methylomirabilaceae* MAG STD1_211 was most closely related to *Methylomirabilis limnetica*, the only genome for this lineage sequenced from a freshwater habitat to date (Fig S9). These two genomes cluster with *Methylomirabilis* sp. 002634395 as a sister clade to *Methylomirabilis* genomes associated with metal-amended bioreactors and ditch sediments (Fig S9). MAG STD1_211 encodes the capacity to reduce nitrate to nitrous oxide in line with all other *Methylomirabilaceae* genomes analyzed (Supplementary Data 5 on the Open Science Framework (OSF) at https://doi.org/10.17605/OSF.IO/6X5ZC[37]). This observation supports that the *Methylomirabilaceae* could participate in the metabolic handoff required by the *Methanoperedenaceae* populations found at a similar abundance at the same location. The potential for syntrophism between the *Methylomirabilaceae* and *Methanoperedenaceae* populations is supported by previous work where these families co-occur in systems where AOM occurred in the presence of oxidized nitrogen[86,87].

The potential to couple methane oxidation to the reduction of oxidized nitrogen species was not limited to MAGs from the *Methanoperedenaceae* and *Methylomirabilaceae* families. The majority of putative methanotrophic MAGs in our dataset (27/31) demonstrated the capacity to reduce nitrite to nitric oxide and a smaller proportion could reduce nitrate to nitrous oxide (8/31) (Fig. 5). These traits spanned multiple families (e.g., *Acetobacteraceae*, *Methylacidiphilaceae*, *Methylococcaceae*, *Methylomirabilaceae*, *Methylomonadaceae*, *Mycobacteriaceae*, and *Nevskiaceae*) (Fig. 5). Although the methanotrophic capacity of members from the *Acetobacteraceae* and *Nevskiaceae* has yet to be experimentally confirmed, these MAGs (i.e., STB_66, STC_13, and STE_114) displayed the capacity to reduce thiosulfate as a potential electron acceptor that could be coupled to methane oxidation (Fig. 5). The majority of putative methanotrophic MAGs (24/31)

also displayed high completion for the high affinity complex IV capable of scavenging nanomolar levels of oxygen[88–90], and a similar proportion (26/31) displayed high completion for the low affinity complex IV (Fig. 5). These observations support that pMMO and sMMO-bearing methanotrophs in leachate have versatile redox metabolisms that can help them survive in landfill habitats separated from the atmosphere.

These adaptations could allow methanotrophs to generate energy in the absence of oxygen as a terminal electron acceptor while also allowing methanotrophs to reserve trace amounts of oxygen to support methane oxidation via the pMMO and/or sMMO pathways. Similar explanations have been put forward for observations where members of the *Methylomonadaceae* were abundant in anoxic aquatic habitats associated with high rates of methane oxidation separate from oxic-anoxic interfaces where aerobic methane oxidation is typically thought to occur[91,92]. Notably, members of the *Methylomonadaceae* have also been detected in studies of landfill cover soils and bioreactors inoculated with leachate subject to oxygen gradients, providing a good proxy for the range of redox conditions considered in our study[10,24,26]. These observations and the widespread detection of *Methylomonadaceae* in leachate frame members of this family as key players capable of limiting methane emissions over a wide range of redox potentials in landfills.

### Phylogenomic analyses of families with no known capacity for methanotrophy

The detection of putative methanotrophs in the *Nevskiaceae, Acetobacteraceae*, which are families lacking in vivo evidence of methanotrophy, and the *Mycobacteriaceae*, which displayed conflicting physiological evidence surrounding the capacity for methanotrophy until very recently[93], prompted us to examine whether the methanotrophic potential observed for these genomes was unique to the landfill-derived populations or more widespread in each lineage. Phylogenomic analyses revealed that multiple genomes spread throughout each family's tree carried the genes coding for complete pMMO and/or sMMO complexes, suggesting the capacity for methanotrophy has been acquired on several occasions within each lineage (see Supplementary Note 3 for detailed descriptions; Figs. S10, S11, S12). Many genomes also displayed near-complete sMMO complexes but consistently lacked the *mmoZ* gene, possibly due to the *mmoZ* gene being prone to divergence as seen in the genomes of known methanotrophs[94] (Figs. S10, S11, S12). Outside of the *Mycobacterium holsaticum* genome associated with human sputum, almost all putative methanotrophs identified across the three lineages occurred in aquatic, terrestrial, and sediment habitats where methane oxidation occurs[28,73].

These observations suggest that many of these lineages are potentially overlooked as methane oxidizers in these habitats. We highlight the *Mycobacteriaceae* as a case study in Supplementary Note 3 to place our phylogenomic analyses in the context of the recent discovery of methanotrophy in an isolate from this family, which clarifies the previous conflicting reports concerning methane oxidation in representatives of this family[93]. In the future, it will be important to take advantage of the fact that many of these strains, in the *Mycobacteriaceae* but also from the other lineages profiled here, exist in culture collections, providing an opportunity to validate their predicted methane oxidation capacity.

## Conclusions

This study provides a historical perspective on biogeochemical succession and the resultant shifts in methane cycling guilds over decades of MSW ageing. Our findings show that geochemical monitoring captures the major processes happening in the landfill but lacks nuance from microbiological information essential to predicting methane's fate in a landfill.

Our metagenomic analyses showed that newer landfill habitats support more diverse microbial communities compared to older landfill habitats where the dominant guilds shift from anaerobic fermentative organisms, to methanogens, to autotrophs with versatile redox metabolisms as waste ages. Methanogens displayed generally low abundance in all but one landfill cell and their community structure in ageing MSW seems to be controlled by the variety and availability of substrates that can support methane production, as well as oxygen infiltration inhibiting methanogenesis. Methanotrophs had a more restricted distribution in terms of how many landfill cells they could be detected in and fewer families with confirmed capacity for methanotrophy were detected compared to families with known methanogens. When present, methanotrophs tended to occur at slightly lower abundance than methanogens. We observed the capacity for methanotrophy across a broad range of metabolisms that likely reflect the steep redox gradients encountered in landfills. The widespread adaptations observed in central redox metabolisms suggest that methanotrophy, even via oxygen-requiring pathways, is important to consider in anoxic landfill habitats.

One of the most exciting findings that emerged from our metagenomic analyses is that pathways and microbial taxa predicted to be involved in the anaerobic oxidation of methane are more diverse than previously described. The abundance of anaerobic methane oxidizers at location D1 among methane cycling microorganisms raises important questions as to what variables result in a habitat favouring anaerobic vs aerobic pathways for methane oxidation. Identifying these variables will be crucial for developing biostimulation strategies that allow landfills to function as giant anaerobic methane-oxidizing bioreactors once methane production has dropped below sustainable bioenergy generation levels. Our work indicates that it is important to dig below cover soils, into the anoxic habitats that dominate landfills, to expand the current concept of the niches and diversity of microbial taxa that contribute to methane oxidation in MSW. Although physiological experiments are required to confirm the methane oxidation capacity in the three different families containing novel putative methanotrophs, this discovery reinforces how the unique microbial communities in landfills can help us better understand biogeochemical cycles across different habitats.

Expanding the known diversity of methane-cycling microorganisms is crucial for improving biogeochemical models that can be used to manage methane emissions in landfills. Such models could increase the effectiveness of waste diversion programs, identify substrate amendments for optimal methane production and recovery for landfills, or limit methane emissions based on waste composition for landfills with higher emission profiles. Our study advocates for emphasizing the biological dimension of landfill lifecycle models so that these tools are not only used for monitoring but also actively mitigating the negative environmental impacts of MSW degradation.

## Methods
### Site description

The site sampled in this study is a sanitary landfill located in the northeastern United States (anonymity by request of site management). The site is equipped with leachate collection and biogas capture systems. MSW at the site is housed in six landfill cells, referred to herein as A, B, C, D, E, and F, which had operated in succession for 39 years at the time of leachate sampling in February 2019 (please refer to Table S1 throughout this section). A is the oldest cell (receiving waste from 1980 to 1982) and F is the youngest, receiving waste as of 2014. Landfill cells A, B, and C are closed and completely capped from receiving waste. Cells D, E, and F remain partially capped due to the implementation of a phased landfilling approach that will eventually create one contiguous landfill cell. The practice of leachate recirculation has shifted over time as MSW management strategies at the landfill have changed, such that leachate from any landfill cell could be

recirculated through the parts of the landfill actively receiving waste. At the time of our sampling, leachate recirculation across the entire landfill site had stopped, which would limit the capacity of older landfill cells to affect the leachate geochemistry of newer ones receiving waste. To facilitate comparisons within such a heterogeneous system, we organized our geochemical analyses around the most active filling periods for each landfill cell.

A total of 8 samples were obtained from landfill cells as part of this campaign [referred to as A, B, C, D1, D2, E, F1, and F2]. Samples D1 and D2 denote leachate collected from two different wells associated with different parts of landfill cell D. These two samples are associated with different drainage areas in cell D that began receiving waste in 1993 and 1995, respectively. The drainage area associated with sample D1 receives leachate from cell D, but also from a valley between cells C and D where the leachate from both cells mixes and where gas is monitored via dedicated sampling ports. Samples F1 and F2 denote leachate collected from two different locations in cell F, which began receiving waste in 2014 and 2016, respectively. Cell F was actively receiving waste at the time of sampling, as were parts of cell D where the cap was removed from areas of cell D that abutted with cell F. Cell E was largely capped at the time of sampling except for the southeast face which is comprised of soil and vegetative cover. Despite these connections, distinct trends in leachate geochemistry were observed at the local scale for each landfill cell. We have used the age of MSW to ground our classification of each cell into the biogeochemical phases of a landfill lifecycle.

### Leachate collection and geochemical analyses

Leachate sampling involved purging wells of standing liquid prior to using a peristaltic pump to recover 1 L of leachate in sterile Nalgene bottles. Samples were stored on ice prior to filtration through a 0.22 μm pore size sterivex filter (Millipore Sigma, Burlington, MA). Leachate filtration was performed the same day as sampling and filters were stored at −80 °C until processed for DNA.

Monitoring records for leachate geochemistry were kindly provided by site owners dating back to 1983 (i.e., 36 years of records). Additional gas flaring data associated with each landfill cell were also provided for the years 2018 and 2019 to capture historical seasonal variation in gas production over a time frame relevant to the sampling expedition. All geochemical measurements were conducted by Brickhouse Environmental consultants using a standardized methodology (not provided). Select variables, namely BOD, COD, pH, redox potential (ORP), the concentrations of organic acids (i.e., acetic acid, butyric acid, isobutyric acid, propionic acid, and valeric acid), bicarbonate (i.e., a proxy for dissolved inorganic carbon), volume of gas flared, and the composition of the gas flared were used to classify each landfill cell to a specific phase (i.e., phase 1 to 5) in the landfill conceptual model.

### DNA extraction, metagenomic sequencing, and genome assembly

DNA was extracted from biomass on filters using the PowerSoil DNA Isolation Kit (Qiagen) using the manufacturer's protocol with the exception that diced filters were added to the bead tube in place of soil. Extracted DNA was evaluated for quality using the NanoDrop 1000 (Thermo Scientific, Waltham, MA) and quantified using the Qubit fluorometric method (Thermo Scientific, Waltham, MA) following the manufacturer's protocol.

Extracted DNA for all samples was sent to The Center for Applied Genomics (Toronto, Canada) for shotgun metagenome sequencing using an Illumina HiSeq platform with paired 2 × 150 bp reads (Illumina, San Diego, CA). Metagenomic reads were quality trimmed using bbduk in the BBTools suite (https://sourceforge.net/projects/bbmap/) and Sickle v1.33[95]. Post-QC, total sequencing ranged from 22.9 (cell E) to 35.1 Gbp (cell F1) (Table S2). Reads were subsequently assembled into scaffolds using SPAdes3 v.3.15.5[96] using -meta and kmers set to

33,55,77,99, and 127. Only scaffolds greater than or equal to 2.5 kbp in length were further analyzed. Metagenomic reads were mapped separately to each curated scaffold assembly using Bowtie2 v2.3.4.1[97].

Scaffolds from a single metagenome were binned using three binning algorithms: CONCOCT v0.4.0, MaxBin2 v2.2.6, and MetaBAT2 v2.12.1[98–100]. The resulting bins were dereplicated for each landfill leachate sample and scored in a consensus-based manner using DAS Tool v1.1.1[101]. To assess bin quality, DAS Tool-processed bins were used as input in CheckM v1.0.13[102], which yielded 1,881 metagenome-assembled-genomes (MAGs) with >70% completion and <10% contamination retained for further analyses [full completion/contamination statistics for MAGs presented in Supplementary Data 6 on the Open Science Framework (OSF) at https://doi.org/10.17605/OSF.IO/6X5ZC[37]]. The retained MAGs captured between 6.7 and 18.7 Gbp of post-QC sequence data (Table S2).

Mean coverage for a MAG was used as the basis for relative abundance calculations to compare microbial communities across the landfill. Mean coverage was calculated by taking the mean coverage values reported for unique scaffold identifiers within a given genome bin. The distribution of the mean coverage values obtained for each MAG is summarized in Fig S13 and the individual mean coverage and relative abundance associated with each MAG can be found in Supplementary Data 1 on the Open Science Framework (OSF) at https://doi.org/10.17605/OSF.IO/6X5ZC[37]. These mean coverage values were subsequently summed together for each site to provide the denominator that would be used to calculate relative abundance. Relative abundance calculations for individual MAGs were calculated by dividing the mean coverage of an individual MAG by the summed mean coverage of the associated sampling site. Relative abundance at the phylum and family levels was calculated by summing the mean coverage for all MAGs classified at a given taxonomic level. Values were converted to percentages by multiplying by 100.

### Genome taxonomy, annotation, and metabolic summaries

Taxonomy was assigned to MAGs using the Genome Taxonomy Database Toolkit application (GTDB-tk) r89 available on DOE-KBase[103]. MAGs were annotated using the Distilled and Refined Annotation of Metabolism (DRAM) tool v1.0 with default parameters[104] but omitting the use of the KEGG and UniRef90 databases for initial annotation of all 1,881 MAGs. In specific cases where <20 genomes from a specific lineage needed to be characterized in additional detail, the UniRef90 database was applied to test whether it improved annotations for key pathways (see Supplementary Data 4, 5, and 6 on the Open Science Framework (OSF) at https://doi.org/10.17605/OSF.IO/6X5ZC[37]). DRAM generates an annotation file that was used alongside the product file and taxonomy data to identify putative methanogens and methanotrophs through additional data manipulation in R v4.2.2 (see Supplementary Data 1 and 7 on the Open Science Framework (OSF) at https://doi.org/10.17605/OSF.IO/6X5ZC[37], associated R code used to produce all figures can be found at https://github.com/carleton-envbiotech/Methane_metagenomics[105]).

### Overview of microbial community

Relative abundance values calculated at the phylum and family level were used to conduct beta diversity analyses and summarize major differences in the most abundant taxa relative to the methane cycling guilds across the landfill. Non-metric multidimensional scaling (NMDS) analysis using the Bray-Curtis dissimilarity matrix was used to assess beta diversity using relative abundance at the family level. NMDS analysis was carried out using the 'vegan' v2.6-4 package in R v4.2.2 and subsequently plotted alongside the total numbers of MAGs recovered from each sample to give a sense of alpha diversity using 'ggplot2' v3.4.2. Relative abundance heatmaps were also generated at the phylum and family level with 0.1 and 5 % cutoff values. All R code used to produce data visualizations can be found at: https://github.com/

carleton-envbiotech/Methane_metagenomics[105] and Supplementary Data 1 can be found on the Open Science Framework (OSF) at https://doi.org/10.17605/OSF.IO/6X5ZC[37].

## Methane cycling microbial community analyses

DRAM's product file and GTDB taxonomic classifications were used to identify putative methanogen MAGs by verifying whether MAGs possessed the *mcrA* gene or a 75 % complete pathway for hydrogenotrophic methanogenesis (equivalent to 6/8 steps being present in the "Methanogenesis, $CO_2$ => methane" pathway output by DRAM) [Supplementary Data 1 on the Open Science Framework (OSF) at https://doi.org/10.17605/OSF.IO/6X5ZC[37]]. MAGs classified to methanogenic lineages but lacking the *mcrA* gene had their annotations verified manually for other genes from the *mcr* operon prior to further analyses [Supplementary Data 7 on the Open Science Framework (OSF) at https://doi.org/10.17605/OSF.IO/6X5ZC[37]]. MAGs that had high completion for the hydrogenotrophic methanogenesis pathway that completely lacked *mcr* genes are discussed separately in Supplementary Note 4. We used the definitions and mechanisms summarized in an authoritative review on methanogens to develop additional rules to classify putative methanogen MAGs as being strictly hydrogenotrophic, acetoclastic, or methylotrophic [Ref. 63 and references therein].

Strictly hydrogenotrophic methanogens can only produce methane by reducing carbon dioxide using hydrogen as an electron donor. MAGs identified as strictly hydrogenotrophic methanogens must have the *mcrA* gene and/or >75% completion of the hydrogenotrophic methanogenesis pathway. MAGs that lacked the *mcrA* gene but displayed high completion for the hydrogenotrophic pathway were further investigated to verify they were taxonomically classified to lineages with known methanogens and whether additional genes from the *mcr* operon were present. MAGs were included in analyses if they harboured any of the *mcrBCDG* genes found in the *mcr* operon. MAGs classified as strictly hydrogenotrophic methanogens also needed to lack the carbon monoxide dehydrogenase/acetyl-CoA synthase complex (CODH/ACS), denoted as "Acetyl-CoA pathway, $CO_2$ => Acetyl-CoA" in DRAM's output. The completion of this pathway is determined based on the presence of genes catalyzing reversible redox transformations between carbon monoxide and carbon dioxide, and methyl group transfers between the coenzyme M precursor tetrahydromethanopterin ($H_4$MPT) and acetyl-CoA[106,107]. This enzyme complex is a hallmark of acetoclastic methanogenesis but is also thought to support autotrophic carbon fixation in archaea bearing near complete hydrogenotrophic methanogenesis pathways that lack the *mcr* operon[107].

Acetoclastic methanogens convert acetate to acetyl-CoA, which subsequently undergoes dismutation to produce carbon dioxide and a methyl group. The carbon dioxide can be further converted to methane using the hydrogenotrophic pathway whereas the methyl group supplied from acetyl-CoA goes towards forming the precursor to coenzyme M, $H_4$MPT-$CH_3$. MAGs identified as putative acetoclastic methanogens needed to have the *mcrA* gene and/or genes coding for enzymes capable of converting acetate to acetyl-CoA. The potential to convert acetate to acetyl-CoA was evaluated based on the presence of genes coding for the acetyl-CoA synthetase alone (labelled as "Acetate pt 1" in DRAM's output), or the acetate kinase and acetyltransferase together (labelled as "Acetate pt 2 and 3", respectively). MAGs identified as putative acetoclastic methanogens also required a > 50% complete CODH/ACS pathway. Given that CODH can further oxidize carbon monoxide to carbon dioxide, MAGs identified as acetoclastic methanogens also required >75% completion for the hydrogenotrophic methanogenesis pathway. The annotation of MAGs displaying high completion of the hydrogenotrophic pathway alongside biomarker genes to convert acetate to acetyl-CoA but lacking the *mcrA* gene were examined to determine whether other

genes in the *mcr* operon were present as a condition to being included in the dataset.

Methylotrophic methanogens can produce methane using methylated compounds. Our categorization in this instance does not distinguish between the $H_2$-dependent methylotrophic methanogens that oxidize methylated compounds to methane and methylotrophic methanogens that rely on the disproportionation of methylated compounds to form methane. This is to facilitate metabolic data interpretation. MAGs identified as putative methylotrophic methanogens needed to have the *mcrA* gene present alongside biomarker genes associated with the methyltransferase enzymes specific to methanol, trimethylamine, dimethylamine, and/or methylamine output by DRAM. DRAM does not output the presence of methylthiol transferases in the product file by default. In cases where methylthiol transferases were suspected as a potential pathway to support methanogenesis, the presence/absence of the *mtsA* and *mtaA* genes was manually verified in the annotation file for select genomes[50]. MAGs identified as methylotrophic methanogens were required to lack the CODH/ACS complex and display low completion (<50%) of the hydrogenotrophic methanogenesis pathway so that only methyl-bearing substrates could potentially support methane production.

We also included a category of broad substrate methanogenesis. This category encompasses MAGs bearing the *mcrA* gene and the potential to access an array of substrates including inorganic carbon, acetate, methanol, and amine-bearing molecules alongside a > 75% complete hydrogenotrophic pathway to produce methane.

MAGs for putative aerobic methanotrophs were identified based on the presence of the *pmoA* and/or *mmoX* genes coding for key subunits in the pMMO and sMMO enzyme complexes in DRAM's product file [see Supplementary Data 1 on the Open Science Framework (OSF) at https://doi.org/10.17605/OSF.IO/6X5ZC[37]]. MAGs taxonomically assigned to ANME were manually identified and reclassified as predicted methanotrophs. In cases where putative methanotrophs were taxonomically classified to families with no previous record of methanotrophy, annotations were manually verified to ensure multiple *pmo* and *mmo* genes occurred on scaffolds with minimum lengths of ~3000 bp prior to subsequent analyses [see Supplementary Data 7 on the Open Science Framework (OSF) at https://doi.org/10.17605/OSF.IO/6X5ZC[37]].

## Compilation of presence/absence data for methanogenic taxa from landfill studies

To compare the occurrence of methanogenic and methanotrophic taxa in this study to previous research, a meta-analysis was conducted for a select number of studies examining landfill microbial communities. The main criterion for inclusion was that these studies examined microbial communities in situ in landfills. Studies that sampled landfills to inoculate enrichment cultures were also considered but data was only collected if the original environmental sample was sequenced, and only that original environmental sample was used in the meta-analysis. Exceptions were made for temporal studies that did not apply selective forces to enrich for specific microbial guilds, but monitored the succession of microbial community associated with solid waste or leachate under conditions that support waste degradation in situ[15,16,18].

Presence/absence was determined by first examining the data presented in figures and tables in the published versions of articles. In specific cases where articles cited accessible supporting information, these data were also incorporated into the analyses. A liberal approach was taken for determining presence/absence. Specific taxa reported in the articles were recorded as being present. For amplicon sequencing surveys, which comprised the bulk of the data compiled, any relative abundance >0% for a given taxa was deemed sufficient to indicate that this taxon was present. For studies that used patterns in restriction fragment length polymorphism or closest relative matches to identify the taxa present, the name of the closest relative was recorded to

indicate a taxon was present. For metagenomic studies, the taxa names associated with metagenome-assembled-genomes or biomarker genes used to assess abundance were recorded as those taxa being present. In instances where specific microarrays or fluorescent in situ hybridization were employed, the species names reported by the authors in the articles were taken as evidence of those taxa being present.

From all compiled data, the deepest level of taxonomic classification presented was recorded [see Supplementary Data 2 on the Open Science Framework (OSF) at https://doi.org/10.17605/OSF.IO/6X5ZC[37]]. Given that our study focussed on comparing microbial communities at the family level, studies that did not provide taxonomic classification to the family-level or deeper were discarded from further analyses. Our study used GTDB release 89 as the database for taxonomic classification[103] whereas the data compiled from the literature relied on a variety of databases for 16S rRNA genes over several years (e.g., Greengenes, SILVA, NCBI)[108–110]. To ensure consistent naming between the taxa compiled from the literature and our study, species or genus names compiled from the literature were manually searched in GTDB and the GTDB name was compared to the NCBI name for searches that provided hits. The naming history of the taxon was also manually verified, such that a list of rules was developed to link older species, genera, and family names to the naming convention for families in GTDB release 89 (e.g., a name commonly reported in the literature was the family *Methanosaetaceae*, which is now *Methanotrichaceae* in GTDB). Conversions identifying the current family naming convention based on species, genera, and family names originally reported have all been reported in the R code that was used to visualize this data with comments linking the first occurrence of this name to a specific GTDB release (available under the directory "Methanogen_metaanalyses" via https://github.com/carleton-envbiotech/Methane_metagenomics[105]).

### Acetate cycling community analyses

Working from DRAM's product file, we developed four potential categories to capture a broad range of potential pathways for acetate production[111]. Category (i) included MAGs with minimum 6/7 steps for the Wood-Ljungdahl (WL) pathway (equivalent to >85% completion) and the genes coding for the phosphotransacetylase (SCFA and alcohol conversions: acetate pt. 1) and acetate kinase (SCFA and alcohol conversions: acetate pt. 2). This classification was designed to capture the methyl branch of the WL pathway. Category (ii) included MAGs with 2/2 steps for the carbon monoxide dehydrogenase/Acetyl-CoA synthase (CODH/ACS) and genes coding for the phosphotransacetylase and acetate kinase. This classification was designed to capture the carbonyl branch of the WL pathway. Category (iii) included MAGs possessing 6/7 steps of the WL pathway but lacking phosphotransacetylase and acetate kinase in the initial annotations. This classification was designed to flag potential MAGs that required further investigation into their acetogenic potential. Category (iv) included MAGs with genes coding for phosphotransacetylase and acetate kinase. This is a broad definition designed to capture MAGs that can produce acetate from acetyl-CoA without the proton-reducing steps associated with the WL and CODH/ACS pathways.

### Phylogenomic analyses

All phylogenomic analyses were conducted with GToTree v1.5.38[112], which references the GTDB release 202 taxonomic identifiers[103,113] when retrieving publicly available genomes. GToTree was used to collect representative genomes within a lineage of interest and related lineages to build outgroups.

To visualize the distribution of methane oxidation marker genes, Pfam identifiers for the pMMO and sMMO pathways were supplied to GToTree. The metadata file output by GToTree with *pmo* and *mmo* gene counts was subsequently used to establish strict criteria for identifying genomes as encoding the pMMO, sMMO, or pMMO and sMMO pathways.

Categories were overlaid onto phylogenetic trees in R v4.1.2 using the packages 'ggtree' v3.0.4[114,115] and 'treeio' v1.16.2[116]. Isolation sources of genomes were manually retrieved using the NCBI biosample number associated with each genome and overlaid onto trees when pertinent. Accession numbers from GToTree were supplied to the 'bit' v1.8.53 package to download genomes, which were included as input for metabolic models produced using DRAM. Examples of R notebooks containing the code used to analyze each lineage of interest are provided alongside the input data required to reproduce these analyses at https://github.com/carleton-envbiotech/Methane_metagenomics[105].

### Reporting summary

Further information on research design is available in the Nature Portfolio Reporting Summary linked to this article.

## Data availability

All data necessary to interpret, verify, and extend the research presented in this article are publicly available. The sequencing data generated in this study have been deposited to NCBI under BioProject PRJNA900590. Within this BioProject, the raw reads files are available on the SRA database, under Biosamples SAMN31696084 – SAMN31696092. The 1,892 MAGs have been deposited to the WGS database under accessions SAMN32731718 – SAMN32731810 (STA), SAMN32731811 – SAMN32731998 (STB), SAMN32733587 – SAMN32733720 (STC), SAMN32734194 – SAMN32734413 (STD1), SAMN32734415 – SAMN32734683 (STD2), SAMN32734737 – SAMN32734946 (STE), SAMN32737191 – SAMN32737484 (STF1), SAMN32737485 – SAMN32737723 (STF2). The geochemical monitoring raw data generated in this study are provided in the Source Data file hosted on the Open Science Framework under https://doi.org/10.17605/OSF.IO/6X5ZC[37] – provider identification is held anonymous at the request of the landfill site engineers. The processed annotation data are provided in the Supplementary Data Files 1-7 which are hosted on the Open Science Framework under https://doi.org/10.17605/OSF.IO/6X5ZC[37]. Source data are provided with this paper.

## Code availability

The input DRAM annotation and product data and accompanying bash and R code used for the analyses in this manuscript has been provided as Supplementary Data for download via https://github.com/carleton-envbiotech/Methane_metagenomics[105].

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

## Acknowledgements

We are sincerely grateful to the landfill site management and their contracted consulting company (anonymity by request) for site access, aid with sampling, and provision of detailed monitoring records. We thank Dr. Jennifer Biddle and her lab group for hosting our team for sample processing prior to shipment. Thanks also to Ms. Rebecca Co and Ms. Alexandra Sauk for help with sampling. This work was supported by an NSERC Discovery Grant (2016-03686) to L.A.H. L.A.H. was supported by a Tier II Canada Research Chair. D.S.G. was supported by an NSERC Banting Postdoctoral Fellowship. N.A.G. was supported by an NSERC CGS-M, an Ontario Graduate Scholarship, an NSERC PGS-D, and a W.S. Rickert Graduate Student Fellowship from the University of Waterloo. We acknowledge the Waterloo Center for Microbial Research for financial support towards open access.

## Author contributions

N.A.G. and L.A.H. contributed to field work in this study. D.S.G. contributed geochemical analyses from records provided by site owners. D.S.G., N.A.G., and L.A.H., contributed to genome assembly and bioinformatics analyses in this study. D.S.G. contributed to the code required for data analyses in this study. D.S.G. and L.A.H. contributed to writing the manuscript.

## Competing interests

The authors declare no competing interests.
