## [Peer Review File · Nature Communications]

Reviewers' Comments:

Reviewer #1:

Remarks to the Author:

Summary

The authors describe the methane metabolising communities from a landfill site that has been in operation for approximately 39 years. They sample the waste and gas that comes from different parts of the landfill to show that there are successional differences in the microbial community dependent on the time since certain locations within the landfill were created. They describe in great detail methanogens and their inferred metabolism along with methanotrophs and their inferred metabolism.

General comments

This manuscript is a metagenomic survey of the methane metabolising microorganisms present in several different location and is well written. The discussion appears to focus on the metabolism of each of the lineages associated with methane metabolism, while methane is an important part of the landfill gases and those microorganisms that contribute to this methane production are also important.

The bacterial population is also important in this context; therefore, it needs to be analysed in this context of nutrient cycling to show how the materials, presumably carbohydrates are metabolised. For these reasons they need to be included in the analyses of the microbial community in the main document to show how the carbohydrates are metabolised before being formed into methane (as many of the MAGs have been generated as per File 1).

This appears to be an important step in the understanding the metabolises that are formed by these microorganisms to then feed into methanogens. Seeing as the MAGs have already been generated can the analyses of the total bacterial community be done to show relationships with those methane-forming and consuming microorganisms? It would be good to have done ordination analysis of the bacterial and archaeal communities to show how important they are under certain conditions (i.e. different cells). This seems relatively easy to achieve given that there is much metadata that is given in the manuscript and will give a better understanding of the microbial community and what drives it (especially for methanogens and mtethanotrophs)

Also, there appears to be anomalies that the authors talk about at several points throughout the manuscript. These anomalies could be eliminated if replication through temporal sampling and analysis was performed. If the samples of D1 and F1 are taken out as these appear to be slightly different sampling sites to all the other cells, this doesn't leave a t lot of samples to understand how similar the community changes over short term and how different the communities especially as the recycling of effluent from older cells into newer cells which would likely influence the other cells.

Isotopic data of the methane, carbon dioxide and water would also bring the data from inference of metabolism and show which methane metabolising microorganisms are predominant.

Also, authors talk about activity without getting into measurement of microbial density or microbial activity as measured by molecular techniques such as metatranscriptomics. I would have liked to compare the levels of these microbial community, methanogens and methanotrophs to how they vary between sites, but this is difficult with the layout of the relative coverage metric which is only useful for comparison within a sampling site.

Also, it would have been good to see the data presented not only as abundances of each MAG but lineages where MAG abundances are summed. This would give a better analysis of the community for understanding the changes that are seen in both the bacterial and archaeal lineages.

The mean coverages are a proxy and are valid for showing which are enriched, but these abundances are hard to read in the context of the community in Figure 2. Especially as the mean

and median coverages of MAGs change between each site dependent on how much data is assembled into MAGs (also there is no data about how much GB of sequence data was produced and how much was assembled into MAGs), therefore it makes any sort of comparison between samples difficult. Why cannot a heat map of these community members relative to its abundance in the whole community be given? It would be more meaningful to put the methanogen/methanotroph community in context of the whole community not just what has assembled (which is likely to be variable between samples). Can MAGs in Figure 2 of the same lineage be combined and incorporated into a heatmap to make it easier to compare between cell sites?

How does the material that the microbial community feeds on over time changes over time? Over a 40-year period society has changed what is going to landfill and this could influence the differences in the microbial communities seen here. how do the authors take this into account? How do plastics (likely more plastics these days) and metals contribute to the microbial metabolism in these environments? Even though MSW is heterogenous, can an average composition of material going to landfill be provided? As this composition will likely influence the microbial communities and potentially lead to differences

After going through all the sampling sites this manuscript has a feel of a list of individual case studies. There appears to be lots of inference about what these methane metabolising microorganisms are performing with respect to their function in the environment. Also, that have the commonality of being associated with the same landfill site with the rigour required to make sure that trends and anomalies that are in this field study site are not replicated to remove the problem. This needs to be rectified by sampling more sites across the landfill and on a timescale that is appropriate for these sites.

Methanogen metabolism types are incorrect in several instances and consequently the arguments for these need to be reworked in several instances throughout the manuscript; *Methanosarcina barkeri* might have the pathway for H₂/CO₂ reduction but might not utilise it; Methanomethylphilaceae are obligate H₂/methylated compound utilising microorganisms not straight methylotrophic' *Methanoculleus* are not acetoclastic (can assimilate acetate as a carbon source).

The manuscript seems overly long for just the comparison about the metabolism of the methanogens and methanotrophs linked to the different cells within the landfill sites. It would have been good to see a more comprehensive analysis of the microbial community as it is highly likely that the bacterial community is responsible for the substrates the methane metabolising microorganisms are utilising. Also, build on the metagenomic inference data to show how the methane metabolising microorganisms are working on a temporal scale through metatranscriptomics and isotopic data.

Specific comments

Line 27: Does the word anaerobic need to be stated here? to be explicit that methane is produced anaerobically and clue the reader into the fact that landfill sites are anaerobic?

Line 29: from when does this 69% occur? State a time so that it is clear from when to when this increase occurs.

Line 32: What is the 'covering a 39-year timeframe'? Is it sampling over a 39-year time period? Is it a measurement of a landfill that contains waste that has been collected over 39 years? From first reading it is not clear from this sentence. Can this be rephrased?

Line 41: Landfills worldwide or other? Can this be clarified?

Line 41: can this be changed to high-income per capita, or developed or other metric that is more descriptive than just high income? I feel that this needs to be stated to give a standardised measure of income.

Lines 106: to 108: I agree with this statement about methanogens and substrates, however, that is not the only reason to do metagenomics. Another sentence needs to be added in above this one to give a broader application of metagenomics in the context of the landfill microbial communities associated with metabolism in general (much like the antibiotic resistance sentence currently above)?

Lines 132-135: does the leachate recirculation in older parts lead to communities being transposed into newer, therefore influencing the communities of the younger sites? Are there any similarities in the community members? Can this be ruled community swapping be ruled out as an effect?

Lines 140-153: A bit difficult to follow with all the exceptions that are occurring, can a table be produce or a deidentified schematic to show where everything is located?

Lines 169: what is the evidence for oxygen intrusion? Are their oxygen measurements? Can they be measured?

Line 182: is there more gas because it is a bigger cell? Therefore, is the comparison of methane flow rates invalid because of different sized cells?

Lines 208-209: were replicates taken and were follow up samplings undertaken to see if this was an anomaly?

Line 211-213: is there more gas because it is a bigger cell? Therefore, is the comparison invalid because of different sized cells? Is there some way this measurement be standardised?

Lines 225-227: Is this anomaly weakens the case for monitoring at a single time point, as this change can affect the microbial community in ways that might evident until a second sampling. Therefore, a monitoring program is warranted for this study to account of any anomalies that are seen here in F1 and F2 given that they are likely have different sources and the differences seen in the bicarbonate that is mentioned on lines 204-210. The authors even stated that this likely led to an altered microbial community analysis.

Lines 235-241: The difference in methane production in Cell E is described in lines 238-241. Does this then mean that the microbial community is fluctuating and there needs to be more samples taken to identify the microbial community that is also fluctuating at this site?

Lines 244-251: the community composition also needs to be calculated with respect to methanogens in relation to bacteria and the absolute amounts needs to also be calculated given that the authors are making the comparisons across the 5 Cells to say that they are decreasing in their methanogenic activity over time. Also, there is no data about how much GB of sequence data was produced and how much was assembled into MAGs. It would be more meaningful to put the methanogen/methanotroph community in context of the whole community not just what has assembled (which is likely to be variable between samples). This is especially true as on line 257 the authors describe the methanogen community as being low abundance but do not provide any evidence for this community being low.

Line 269: where in the supplemental results? Can the authors point to a specific figure or table?

Line 271: define low abundance, is it low relative abundance compared to what? The community needs to be quantified for these sorts of statements to be made as this is just low relative abundance within the sample, they might be higher absolute abundance than in the other cells. Although low abundance might correlate with low methane levels, they might just be lowly active but high in number. Absolute numbers need to be quantified.

Lines 274-276: this is a bit impossible to measure as the this is probably a mix of different layers within the cell mixing as they come out of the pile, the data in figure 1 are highly variable and likely reflect this point, a combination of different levels of methane and oxygen. Can this sentence be rephrased to reflect this difference?

Lines 282-284: from the description of D1 and D2 in the first section, it is not clear if the sources of these materials are the same hence this discrepancy in the putative methanogenic structures. Can this observation be incorporated here?

Lines 305-308: While Methanotrichaceae are acetoclastic, Methanoregulaceae and Methanocullaceae are not acetoclastic methanogens, these lineages are from the Methanomicrobiales, which are not an acetoclastic lineage. Please rephrase the sentence.

Lines 309-311: Also, some of these methanogens described in this sentence are strict hydrogenotrophs, but looking at figure 2 for section D1 it appears that they are acetoclastic (acetate/methane, pt1) based on this analysis.

Lines 311-312: Again these are not acetoclastic methanogens, they might assimilate acetate as a carbon source through an acetyl-CoA synthase, but are not acetoclastic (maybe this is a flaw in DRAM?). Rephrase this sentence to reflect this fact.

Lines 312-313: Methanomethylphilaceae while methylotrophic, are (at this stage) have a strict requirement for H₂. So are H₂ dependent methylotrophs. Can this sentence be changed to reflect this fact?

Lines 313-316: Although they encode genes for hydrogenotrophic methanogenesis, not all Methanosarcinaceae methanogens utilise H₂/CO₂. An example of this is Methanosarcina barkeri. Can this sentence be rephrased to reflect this fact?

Line 321: samples taken over a greater timescale would alleviate the potential for this sampling to be an anomaly vs are true reflection of the community because of the seeding from the older cells. Can more sampling of this site be done to confirm this?

Lines 362-364: this result suggests that there is a strong need for temporal sampling in this study to understand any anomalies seen across the sites and described in other parts of the manuscript. Can this data be augmented by extra data? As the authors pointed out this needs to be extrapolated with more sampling across the landfill site.

Line 390-391: Can some evidence be produced to show that they produce methane rapidly in these conditions. As conversely, they might not be well suited just generalists so therefore they are found in many environments. Please rephrase to reflect this alternative hypothesis.

Line 397: Are these Methanosarcinaceae utilising H₂/CO₂ or do they just have the pathway? As seen in the Methanosarcina barkeri, has hydrogenotrophic pathway but doesn't utilise it for H₂/CO₂ reduction.

Line 407-409: Methanofastidiosia are currently predicted to produce methane from methane thiol based on reference 35. Rephrase this sentence accordingly.

Lines 410-411: Methanocullaceae are not acetoclastic methanogens, they likely possess an acetyl-CoA synthase to assimilate acetate as a carbon source but are not produce methane from this substrate. This would be ground-breaking if it is proven true. Or alternatively change this sentence and others in the manuscript to reflect this accordingly.

Barret et al. (2013) Identification of Methanoculleus spp. as Active Methanogens during Anoxic Incubations of Swine Manure Storage Tank Samples. Appl Environ Microbiol. 79:424-433

Line 411: what supplemental results? please name a specific figure or table.

Lines 450-456: given that there are both Methanosarcinaceae and Methanotrichaceae present it would be good to describe some of this with respect to the niche of acetate concentration. As Methanotrichaceae (µM), have a lower threshold for acetate than Methanosarcina (mM) and will go to niche differentiation in relation to the landfill cells.

Line 456: this paragraph seems very general and is making inference about the methanogens

MAGs compared to other that have been studied without any analysis of the MAGs in this study. From this list named all MAGs in the environment have stress tolerance genes. Analysis needs to be done to confirm.

Line 462: inferred to utilise methylated thiols, rather than known. Please change this sentence.

Lines 469-471: need to work it into the sentence that these methanogens are obligate H₂/methylated compound utilising methanogens and contribute cycling of both these compounds.

Lines 496-556: it feels like there is a lot of description here to describe the different MAGs that were recovered from the different cell sites. Can this be cut down to something more concise. When combined with the abundances for each it becomes a little bit too much. This would be better spent putting the results in context of the environment.

Lines 595-597: Is this suggested syntrophism supported here in the landfill sites? Do they co-occur in sites and at increasing abundances? Compare them to support this claim, otherwise remove this sentence as it is speculation.

Lines 614-624: Can any of the methanotrophic bacteria (excl. NC-10) be correlated with oxygen concentrations? What are the abundances for Type I and Type II methanotrophs which have different oxygen optima? All these would be useful analysis to put these in context of the environment.

Line 636: despite the absence of mmoZ genes can other potential homologs that could perform the same function be identified to verify these proposed pathways?

Line 658: 'More restricted distribution and diversity' compared to what? It is not clear state this and reference this.

Lines 658-660: It is hard to compare between these because they are not on the same plot anywhere. Please put them together so that comparison can easily be made. Also, total abundances of lineages is preferable rather than just looking comparison of individual genomes as it makes the comparison even more difficult between groupings and different metabolism types.

Lines 725-727: The mean coverages are a proxy and are valid for showing which are enriched, but these abundances are hard to read in the context of the community in Figure 2. Especially as the mean and median coverages of MAGs change between each site dependent on how much data is assembled into MAGs (also there is no data about how much GB of sequence data was produced and how much was assembled into MAGs), therefore it makes any sort of comparison between samples difficult. Why cannot a heat map of these community members relative to its abundance in the whole community be given? It would be more meaningful to put the methanogen/methanotroph community in context of the whole community not just what has assembled (which is likely to be variable between samples).

Reviewer #2:

Remarks to the Author:

The present research investigates the methane cycle with respect to microbial community population.

The introduction provides sufficient background information. However, could consider reducing the length of the text in lines 84-100 and lines 84-91 may precede the last paragraph.

Lines 101-111 are also very general and not provide any information on the subject, whereas the physicochemical characteristics are overlooked.

Lines 125-153 are a part of the methodology, where additional information on the landfill characteristics should be included (amount of waste deposited, climatic conditions, volume of

leachate produced etc).

Lines 138-139, please be more specific when referring to active filling periods, as well as the condition of the landfills during the sampling.

The authors did not consider the value of the simple BOD/COD ratio, as in many cases researchers characterize a leachate sample based on the age on the landfill and not on the actual data. Authors should include a plain table containing the physicochemical characteristics of the raw samples and the respective measurements at the sampling day.

The ammonium nitrogen content and the electrical conductivity are overlooked. Alkalinity is between VFAs. All these crucial measurements are lost in supplementary material graphs.

Did authors extract DNA only once? How many replicate samples were analysed from each site?

Fig 1B does not clearly present sample area D1, whereas oxygen presence in A and B is not adequately discussed and the pH.

Line 248, please correct 'Halobacteriota' (Rinke and co-workers, 2021, A standardized archaeal taxonomy for the Genome Taxonomy Database').

Could authors explain the sharp decrease of pH in cell E?

Please include the main findings of the metagenomic analysis, even at family level, in a comparative table.

The dominance and relative abundance of each phylum and family should be clearly stated, like in line 470 and in methanotrophy results, instead of just the classification.

Why the methanol dehydrogenase gene (mxoF) was not included in the research.

Lines 664-648 are too general and not in compliance with lines 540-544 where reference is made regarding the presence of Methanoperedenaceae MAGs.

The methodology of physicochemical analyses is absent.

The research is very interesting, but most findings are scattered in numerous supplementary figures and tables, which do not help the clarity of the presentation.

Response to Reviewer Comments for Nature Communications Editor

Main points of concern

We thank both Reviewers for their thoughtful review of our manuscript, and the Editor for his continued interest in the paper. Many of the suggested revisions strengthened the paper and we have attempted to streamline the text while adding the additional data analyses requested to improve the manuscript. In response to the Reviewers comments, we have summarized the full microbial community at the site, added detail to the total community metabolic predictions, and made additional efforts to streamline the geochemical analyses in the manuscript.

At this time, we cannot obtain replicated samples, additional time points for the microbial community and geochemistry, metatranscriptomic data, or isotopic ratios. We do not have samples collected that allow these analyses, nor the funding to support new analyses, nor access to the site for temporally disjointed samples to process anew (which would introduce different concerns). Although these are valuable recommendations and would certainly be of interest to us if feasible, they fall outside of the scope of our study, which focussed on providing an unprecedented historical perspective on methane cycling guilds by coupling comprehensive metagenomic surveys to long term geochemical records. The current study stands on its own as an independent body of work that advances the field.

We have indicated the general and specific comments that have been addressed in the newest version of the manuscript below. Edits have been highlighted in the manuscript file to facilitate review.

Comments from Reviewer #1 (Reviewer comments highlighted)

Reviewer 1 made comments about the current understanding of methanogenesis in model organisms that run counter to the putative methanogenic metabolism observed in the landfill. We stand by our objective interpretation of our metabolic data because we think it is important to consider metabolic adaptations the methane cycling guilds exhibit in landfills that may run counter to the current paradigm of methanogenesis based on lab studies of model organisms. Reviewer 1 indicated that these comparisons should be highlighted in specific parts of the manuscript, which we agree would provide additional context in select instances, and we have included information to that point in the revised manuscript.

Summary

The authors describe the methane metabolising communities from a landfill site that has been in operation for approximately 39 years. They sample the waste and gas that comes from different parts of the landfill to show that there are successional differences in the microbial community dependent on the time since certain locations within the landfill were created. They describe in great detail methanogens and their inferred metabolism along with methanotrophs and their inferred metabolism.

General comments

This manuscript is a metagenomic survey of the methane metabolising microorganisms present in several different location and is well written. The discussion appears to focus on the metabolism of each of the lineages associated with methane metabolism, while methane is an important part of the landfill gases and those microorganisms that contribute to this methane production are also important.

The bacterial population is also important in this context; therefore, it needs to be analysed in this context of nutrient cycling to show how the materials, presumably carbohydrates are metabolised. For these reasons they need to be included in the analyses of the microbial community in the main document to show how the carbohydrates are metabolised before being formed into methane (as many of the MAGs have been generated as per File 1).

This appears to be an important step in the understanding the metabolises that are formed by these microorganisms to then feed into methanogens. Seeing as the MAGs have already been generated can the analyses of the total bacterial community be done to show relationships with those methane-forming and consuming microorganisms?

Both reviewers identified the surrounding microbial community as a missing piece of the paper. We have included beta diversity analyses at the family level (new **Fig 2**) in addition to relative abundance heatmaps at the phylum (new **Fig S3**) and family level (new **Fig S4**) for the whole community. The relative abundance data is also placed in context alongside the abundance of methane cycling guilds in the landfill (revised **Fig 3** and a new **Fig S5**).

A full workup of all carbon cycling pathways that could feed into methanogenesis is outside of the scope of this study. We have chosen to ground our metabolic interpretations in methane data and the historical geochemical records, which provide sufficient context for the substrates available to support methanogenesis. To address this point, we have provided an overview of putative acetogen MAGs that could potentially supply substrates to methanogens (new **Fig S6**). Detailed interpretation of these additional results has been moved to the Supplementary Results while a summary of these findings is included in the manuscript that speaks to the potential interactions between acetogens and methanogens at L490-494.

It would be good to have done ordination analysis of the bacterial and archaeal communities to show how important they are under certain conditions (i.e. different cells). This seems relatively easy to achieve given that there is much metadata that is given in the manuscript and will give a better understanding of the microbial community and what drives it (especially for methanogens and methanotrophs)

We have carried out NMDS ordination using relative abundance data at the family level to address this point (new **Fig 2**). Additional sections have been added to the manuscript that provide an overview of differences in community structure and the dominant phyla (new **Fig S3**) and families (new **Fig S4**). Brief metabolic interpretation of the most abundant families has been provided to give additional insights into the biogeochemistry of the landfill. Our capacity to apply predictive ordinations (e.g., canonical correspondence analyses) is limited given the dataset comprises coupled 'omics and geochemical data for 8 data points.

Also, there appears to be anomalies that the authors talk about at several points throughout the manuscript. These anomalies could be eliminated if replication through temporal sampling and analysis was performed. If the samples of D1 and F1 are taken out as these appear to be slightly different sampling sites to all the other cells, this doesn't leave a lot of samples to understand how similar the community changes over short term and how different the communities especially as the recycling of effluent from older cells into newer cells which would likely influence the other cells.

We were logistically limited in our capacity to obtain multiple replicates from each landfill cell for metagenomic sequencing. It is not possible currently to obtain true replicates or extend a temporal analysis. While we agree a temporal analysis of the same landfill cells would be interesting, it is outside of the historical perspective we provide in this manuscript.

Isotopic data of the methane, carbon dioxide and water would also bring the data from inference of metabolism and show which methane metabolising microorganisms are predominant.

Also, authors talk about activity without getting into measurement of microbial density or microbial activity as measured by molecular techniques such as metatranscriptomics. I would have liked to compare the levels of these microbial community, methanogens and methanotrophs to how they vary between sites, but this is difficult with the layout of the relative coverage metric which is only useful for comparison within a sampling site.

These comments request data not provided in the original study, and which are well outside the scope of work presented. They are good suggestions for future work at this site, but it is not possible to overlay isotopic analyses or obtain metatranscriptomic analyses for these samples currently. We have made a comment on the benefit of using such approaches at L479-483.

Also, it would have been good to see the data presented not only as abundances of each MAG but lineages where MAG abundances are summed. This would give a better analysis of the community for understanding the changes that are seen in both the bacterial and archaeal lineages.

See comments above for additions to the manuscript discussing phylum, family, methane cycling guilds, and MAG-level relative abundance data. We have updated **File S1** to include the relative abundance for each MAG.

The mean coverages are a proxy and are valid for showing which are enriched, but these abundances are hard to read in the context of the community in Figure 2. Especially as the mean and median coverages of MAGs change between each site dependent on how much data is assembled into MAGs (also there is no data about how much GB of sequence data was produced and how much was assembled into MAGs), therefore it makes any sort of comparison between samples difficult.

We have provided the total Gb of sequence data produced in this work and amounts assembled into MAGs in **Table S1** and the methods text. We appreciate that without this information it was impossible to know that the datasets were well-balanced and did not require normalization to

make comparisons between sites valid. Coverage data has been substituted for relative abundance data in our methanogen (now **Fig 3**) and methanotroph (now **Fig 4**) heatmaps to facilitate comparisons. Relative abundance data has also been used to summarize whole-community analysis at the phylum and family level in **Fig S3** and **Fig S4**, respectively, in addition to providing an overview of methane cycling guilds in **Fig S5**. Additional details on how relative abundance was calculated from coverage data has been added to the Methods at L851-862.

Why cannot a heat map of these community members relative to its abundance in the whole community be given? It would be more meaningful to put the methanogen/methanotroph community in context of the whole community not just what has assembled (which is likely to be variable between samples). Can MAGs in Figure 2 of the same lineage be combined and incorporated into a heatmap to make it easier to compare between cell sites?

See comments above regarding heatmaps summarizing relative abundance.

How does the material that the microbial community feeds on over time changes over time? Over a 40-year period society has changed what is going to landfill and this could influence the differences in the microbial communities seen here. how do the authors take this into account? How do plastics (likely more plastics these days) and metals contribute to the microbial metabolism in these environments? Even though MSW is heterogenous, can an average composition of material going to landfill be provided? As this composition will likely influence the microbial communities and potentially lead to differences

Changes in the average composition of the composition of MSW cannot be provided beyond discussions highlighting qualitative data with the site owners. This level of granular data is not recorded by site owners. We confirmed that there has not been an organic waste diversion program implemented within the timeframe of this study. Plastics do impact microbial communities, but largely as surfaces for biofilms on the timescales implicated here.

After going through all the sampling sites this manuscript has a feel of a list of individual case studies. There appears to be lots of inference about what these methane metabolising microorganisms are performing with respect to their function in the environment. Also, that have the commonality of being associated with the same landfill site with the rigour required to make sure that trends and anomalies that are in this field study site are not replicated to remove the problem. This needs to be rectified by sampling more sites across the landfill and on a timescale that is appropriate for these sites.

Obtaining replicated samples is not logistically possible currently. Although we agree that a time series would be interesting, our geochemical records indicate that this would need to be carried out on the order of 3-5 years to capture the succession of the landfill lifecycle. This is not possible currently due to logistical limitations.

Methanogen metabolism types are incorrect in several instances and consequently the arguments for these need to be reworked in several instances throughout the manuscript; *Methanosarcina barkeri* might have the pathway for H₂/CO₂ reduction but might not utilise it;

Methanomethylophilaceae are obligate H₂/methylated compound utilising microorganisms not straight methylotrophic. *Methanoculleus* are not acetoclastic (can assimilate acetate as a carbon source).

Metabolic interpretations were carried out based on the genetic annotations obtained for MAGs and subsequently cross-referenced with the literature where required. Methylotrophic methanogenesis is broadly defined here to capture the H₂-dependent methylotrophic methanogens that oxidize methylated compounds to methane and methylotrophic methanogens that rely on the disproportionation of methylated compounds to form methane to streamline metabolic interpretations.

We have added a clarification in the manuscript at L543 to 549 and Supplemental Methods surrounding methylotrophic methanogenesis. We also make specific mention of the H₂-dependent methylotrophic methanogenesis pathways associated with the order *Methanomassiliicoccales*, which contains the family *Methanomethylophilaceae*, in the manuscript. We have verified the additional cases brought up in this comment to ensure our metabolic interpretations align with the data and provide citations to support conflicting information where needed.

We anticipate some discrepancies between genetic repertoires of laboratory model organisms and MAGs for populations in the landfill given they are distinct species. This is important to consider in the context of the *Methanomethylophilaceae* where the current understanding of H₂-dependent methylotrophic methanogenesis stems from experimental work on a handful of model organisms¹⁻⁴. Many of the *Methanomethylophilaceae* MAGs recovered from our landfill site could not be classified to a named genus. We have attempted to strike a balance between acknowledging one of the defining characteristics of the order *Methanomassiliicoccales* while emphasizing the importance of considering the dual contributions methylotrophic methanogens can make to hydrogen and methane cycling at L548.

The manuscript seems overly long for just the comparison about the metabolism of the methanogens and methanotrophs linked to the different cells within the landfill sites. It would have been good to see a more comprehensive analysis of the microbial community as it is highly likely that the bacterial community is responsible for the substrates the methane metabolising microorganisms are utilising. Also, build on the metagenomic inference data to show how the methane metabolising microorganisms are working on a temporal scale through metatranscriptomics and isotopic data.

We have inorganic and organic carbon substrate data to demonstrate the substrates available to methanogens and methanotrophs. As such, we believe a full detailed interpretation of this complex microbial community (~i.e. over 1,800 high quality MAGs) metabolic capacity would not add to the manuscript, and would represent predictions of metabolites generated versus the actual measurements of metabolites presented. We also feel this would not address the concerns surrounding length as this is a substantially more complicated carbon cycling network to describe. Obtaining multiple time points and applying isotope and metatranscriptomic data is outside of the scope of this study.

Specific comments

All specific comments in grey have been addressed in the manuscript except for those in italics where a response to the reviewer has been provided.

Line 27: Does the word anaerobic need to be stated here? to be explicit that methane is produced anaerobically and clue the reader into the fact that landfill sites are anaerobic?

Removed.

Line 29: from when does this 69% occur? State a time so that it is clear from when to when this increase occurs.

We specified that this is based on 2018 estimates in L31.

Line 32: What is the 'covering a 39-year timeframe'? Is it sampling over a 39-year time period? Is it a measurement of a landfill that contains waste that has been collected over 39 years? From first reading it is not clear from this sentence. Can this be rephrased?

This sentence has been changed to say we sampled waste that has been landfilled over a 39 year period in L34-35.

Line 41: Landfills worldwide or other? Can this be clarified?

The qualifier worldwide has been added for clarity here at L46.

Line 41: can this be changed to high-income per capita, or developed or other metric that is more descriptive than just high income? I feel that this needs to be stated to give a standardised measure of income.

GDP per capita data has been included based on the World Bank's definition along the most recent data available for the United States in L48-49.

Lines 106: to 108: I agree with this statement about methanogens and substrates, however, that is not the only reason to do metagenomics. Another sentence needs to be added in above this one to give a broader application of metagenomics in the context of the landfill microbial communities associated with metabolism in general (much like the antibiotic resistance sentence currently above)?

An additional sentence has been added at L106-112 that speaks to examining methane cycling in the broader metabolic context of other carbon cycling pathway and central redox metabolisms in line with the major edits made to the manuscript.

Lines 132-135: does the leachate recirculation in older parts lead to communities being transposed into newer, therefore influencing the communities of the younger sites? Are there any

similarities in the community members? Can this be ruled community swapping be ruled out as an effect?

Leachate recirculation had stopped so this would be limited. We cannot completely rule out microbial communities swapping between landfill cells. Landfill cells generally followed the chronological order expected and outside of landfill cell E, where other geochemical variables may be influencing community composition, we did not make any observations suggesting a disconnect between microbial communities and geochemistry that would suggest community swapping.

Lines 140-153: A bit difficult to follow with all the exceptions that are occurring, can a table be produce or a deidentified schematic to show where everything is located?

This data has been summarized in the new **Table S1**. Site owners have requested we do not use schematics or maps that could be linked back to the site.

Lines 169: what is the evidence for oxygen intrusion? Are their oxygen measurements? Can they be measured?

Oxygen measurements are provided in the gas flaring data in **Fig 1B**.

Line 182: is there more gas because it is a bigger cell? Therefore, is the comparison of methane flow rates invalid because of different sized cells?

This is a valid comment. We do not have acreage data for D1 vs D2, but we do have it for cell D in its entirety. Landfill cells generally increase in size from cell A ~ 9 acres to ~20 for cells C and D and cells E and F are > 30 acres (summarized in **Table S1**). Although this size could influence cumulative methane production, we were given 2 years of gas flow data from flaring vents and do not have the capacity to test how different landfill cell sizes would affect gas flow for the same type of waste being mineralized over the same period. We have made note of landfill size where appropriate in our geochemical data interpretations at L145.

Lines 208-209: were replicates taken and were follow up samplings undertaken to see if this was an anomaly?

See our comments above – no.

Line 211-213: is there more gas because it is a bigger cell? Therefore, is the comparison invalid because of different sized cells? Is there some way this measurement be standardised?

See comment above regarding landfill cell size.

Lines 225-227: Is this anomaly weakens the case for monitoring at a single time point, as this change can affect the microbial community in ways that might evident until a second sampling. Therefore, a monitoring program is warranted for this study to account of any anomalies that are

seen here in F1 and F2 given that they are likely have different sources and the differences seen in the bicarbonate that is mentioned on lines 204-210. The authors even stated that this likely led to an altered microbial community analysis.

Although a monitoring study would be valuable, it falls outside of the scope of this initial environmental, which establishes the baseline for future work.

Lines 235-241: The difference in methane production in Cell E is described in lines 238-241. Does this then mean that the microbial community is fluctuating and there needs to be more samples taken to identify the microbial community that is also fluctuating at this site?

See comment above. There are also complex reasons for gas fluctuations, including development of natural fissures that lead to uncontrolled methane release where gas is diverted from flares, atmospheric pressure changes, and changes in substrate availabilities that would impact methanogenesis but not necessarily the microbial community composition given the timeframe of the fluctuations (methanogens being reasonably persistent).

Lines 244-251: the community composition also needs to be calculated with respect to methanogens in relation to bacteria and the absolute amounts needs to also be calculated given that the authors are making the comparisons across the 5 Cells to say that they are decreasing in their methanogenic activity over time. Also, there is no data about how much GB of sequence data was produced and how much was assembled into MAGs. It would be more meaningful to put the methanogen/methanotroph community in context of the whole community not just what has assembled (which is likely to be variable between samples). This is especially true as on line 257 the authors describe the methanogen community as being low abundance but do not provide any evidence for this community being low.

See response to General Comments on providing amount of Gb sequences and summary data for the whole community composition. The comment about looking at the methane cycling community outside of the MAGs assembled requires clarification. We suspect the Reviewer means analyzing read-level data to capture sequences that did not fall into high quality bins. We chose to use MAGs as our unit of analyses to facilitate metabolic modeling and connecting potential substrates to the methane cycling pathways observed. Given this reviewer's comment suggesting that these observations be connected to carbon cycling across the landfill, we think our choice to use MAGs vs read-level data is appropriate given the goals of our study.

Line 269: where in the supplemental results? Can the authors point to a specific figure or table?

Here we are referring to the rules we used to classify putative methanogen MAGs. Reference to the proper section in the Supplemental Results file has been provided.

Line 271: define low abundance, is it low relative abundance compared to what? The community needs to be quantified for these sorts of statements to be made as this is just low relative abundance within the sample, they might be higher absolute abundance than in the other cells. Although low abundance might correlate with low methane levels, they might just be lowly active but high in number. Absolute numbers need to be quantified.

We have substituted coverage data for relative abundance data and updated interpretations surrounding methane cycling guilds accordingly throughout the manuscript.

Lines 274-276: this is a bit impossible to measure as this is probably a mix of different layers within the cell mixing as they come out of the pile, the data in figure 1 are highly variable and likely reflect this point, a combination of different levels of methane and oxygen. Can this sentence be rephrased to reflect this difference?

We respectfully disagree with this statement. We sampled leachate, which would provide a homogenous sample of heterogeneous layers of MSW geochemistry. Although there is variability observed across the geochemistry data, we focussed on the smoothed and longer-term trends considering the time frame considered in the landfill lifecycle. We think the metabolic evidence aligns with the geochemical prediction to support this statement.

Lines 282-284: from the description of D1 and D2 in the first section, it is not clear if the sources of these materials are the same hence this discrepancy in the putative methanogenic structures. Can this observation be incorporated here?

A comment has been provided speaking to this discrepancy at L370-373.

Lines 305-308: While Methanotrichaceae are acetoclastic, Methanoregulaceae and Methanoculleaceae are not acetoclastic methanogens, these lineages are from the Methanomicrobiales, which are not a acetoclastic lineage. Please rephrase the sentence.

In this instance, we are reporting how these MAGs were classified considering our decision framework because we do not rule out that diverse populations in landfills could exhibit metabolic capacity beyond what has been reported in previous work. We indicate the conflicting information surrounding the *Methanoculleaceae* in L516-518 of the manuscript. We think it is best to present this data objectively based on our classification framework, highlighting discrepancies in our data compared to the current literature.

Lines 309-311: Also, some of these methanogens described in this sentence are strict hydrogenotrophs, but looking at figure 2 for section D1 it appears that are acetoclastic (acetate → methane, pt1) based on this analysis.

These MAGs were discussed as being acetoclastic based on metabolic interpretations instead of previous precedent in the field.

Lines 311-312: Again these are not acetoclastic methanogens, they might assimilate acetate as a carbon source through an acetyl-CoA synthase, but are not acetoclastic (maybe this is a flaw in DRAM?). Rephrase this sentence to reflect this fact.

We have accounted for this in our classification of acetoclastic methanogens wherein putative acetoclastic methanogens must demonstrate >50% completion of the CODH/ACS pathway. We can add that this observation runs counter to physiological experiments, but we think it is

important to present the data as is because it may reflect increased metabolic capacity as an adaptation to landfill habitats.

Lines 312-313: Methanomethylphilaceae while methylotrophic, are (at this stage) have a strict requirement for H₂. So are H₂ dependent methylotrophs. Can this sentence be changed to reflect this fact?

Additional content indicating the H₂-dependent methylotrophic methanogenesis pathways associated with members of the *Methanomassiliicoccales* has been added at L543 to 549.

Lines 313-316: Although they encode genes for hydrogenotrophic methanogenesis, not all Methanosarcinaceae methanogens utilise H₂/CO₂. An example of this Methanosarcina barkeri. Can this sentence be rephrased to reflect this fact?

We are interpreting our data objectively and are mindful of the language used here, which is why we use the word potential. We think that highlighting very potential exception that challenges our classification framework would add unnecessary length to the manuscript, which has already been framed as too long.

Line 321: samples taken over a greater timescale would alleviate the potential for this sampling to be an anomaly vs are true reflection of the community because of the seeding from the older cells. Can more sampling of this site be done to confirm this?

Additional samples cannot be taken at this time.

Lines 362-364: this result suggests that there is a strong need for temporal sampling in this study to understand any anomalies seen across the sites and described in other parts of the manuscript. Can this data be augmented by extra data? As the authors pointed out this needs to be extrapolated with more sampling across the landfill site.

Additional samples cannot be taken to generate more data at this time.

Line 390-391: Can some evidence be produced to show that they produce methane rapidly in these conditions. As conversely, they might not be well suited just generalists so therefore they are found in many environments. Please rephrase to reflect this alternative hypothesis.

This content has been removed through revisions made to the manuscript.

Line 397: Are these Methanosarcinaceae utilising H₂/CO₂ or do they just have the pathway? As such, seen in the Methanosarcina barkeri, has hydrogenotrophic pathway but doesn't utilise it for H₂/CO₂ reduction.

We can only discuss the presence of pathways in this manuscript. Presumably, these pathways are maintained because they are useful under specific conditions in the landfill that may not have been successfully reproduced in past work with *Methanosarcina barkeri*.

Line 407-409: Methanofastidioisa are currently predicted to produce methane from methane thiol based on reference 35. Rephrase this sentence accordingly.

This sentence has been changed to state specifically that methylthiol is the substrate in line with statements provided by authors in the associated citation (see L472).

Lines 410-411: Methanocullaceae are not acetoclastic methanogens, they likely possess an acetyl-CoA synthase to assimilate acetate as a carbon source but are not produce methane from this substrate. This would be ground-breaking if it is proven true. Or alternatively change this sentence and others in the manuscript to reflect this accordingly.

Barret et al. (2013) Identification of Methanoculleus spp. as Active Methanogens during Anoxic Incubations of Swine Manure Storage Tank Samples. Appl Environ Microbiol. 79:424–433

We thank the reviewer for putting this reference to our attention. We have rephrased this section and included the citation to further emphasize the conflicting information surrounding representative of the *Methanocullaceae*. We have also added a sentence that highlights the thoughtful suggestion of this reviewer to apply isotope-based approaches in future work to address the limitations noted in this study (L479-483).

Line 411: what supplemental results? please name a specific figure or table.

Reference to the specific subheader associated with these supplemental results has been provided at L493.

Lines 450-456: given that there are both Methanosarcinaceae and Methanotrichaceae present it would be good to describe some of this with respect to the niche of acetate concentration. As Methanotrichaceae (uM), have a lower threshold for acetate than Methanosarcina (mM) and will go to niche differentiation in relation to the landfill cells.

A sentence has been added at L464-468 offering this potential explanation alongside a citation.

Line 456: this paragraph seems very general and is making inference about the methanogens MAGs compared to other that have been studied without any analysis of the MAGs in this study. From this list named all MAGs in the environment have stress tolerance genes. Analysis needs to be done to confirm.

We originally carried out manual searches of gene annotations to confirm presence of oxidative stress tolerance genes examined in previous work⁵. We felt it was appropriate to cite Lyu et al., which carried out comprehensive genetic inventories of oxidative stress genes considering the length of the manuscript. These data can be found in the annotation files we have provided in File S2 and we have omitted further data summaries to prioritize streamlining the manuscript.

Line 462: inferred to utilise methylated thiols, rather than known. Please change this sentence.

Sentence has been changed.

Lines 469-471: need to work it into the sentence that these methanogens are obligate H₂/methylated compound utilising methanogens and contribute cycling of both these compounds.

See comments above about H₂-dependent methylotrophic methanogenesis.

Lines 496-556: it feels like there is a lot of description here to describe the different MAGs that were recovered from the different cell sites. Can this be cut down to something more concise. When combined with the abundances for each it becomes a little bit too much. This would be better spent putting the results in context of the environment.

We have attempted to streamline interpretation of methane cycling communities as much as possible throughout the manuscript.

Lines 595-597: Is this suggested syntrophism supported here in the landfill sites? Do they co-occur in sites and at increasing abundances? Compare them to support this claim, otherwise remove this sentence as it is speculation.

We were careful with our language here to focus on the potential for this syntrophism to be supported because this is one of the first reports of these families co-occurring in the landfill. We think the language used is appropriate considering the references provided.

Lines 614-624: Can any of the methanotrophic bacteria (excl. NC-10) be correlated with oxygen concentrations? What are the abundances for Type I and Type II methanotrophs which have different oxygen optima? All these would be useful analysis to put these in context of the environment.

We are reticent to characterize MAGs along the dimension of Type I vs Type II because of large-scale genetic surveys of the *pmoA* gene which have revealed methanotrophs from a broad range of habitats exhibit more diverse lifestyles than can be captured by this dichotomy (Knief 2015, ref 27 in the original manuscript, now ref 28 in the new version). The qualitative overview we provide of where methanotrophs occur based on where oxygen was detected in the landfill is sufficient to support our analyses.

*Line 636: despite the absence of *mmoZ* genes can other potential homologs that could perform the same function be identified to verify these proposed pathways?*

We chose to interpret our results conservatively in this case, focussing on MAGs with the full complement of *mmo* genes annotated. The *mmoZ* gene codes for the gamma subunit of the soluble methane monooxygenase complex. It is possible that a homolog could fulfill this function but outside of some speculation, we feel that this investigation would be outside of the scope of this study.

Line 658: 'More restricted distribution and diversity' compared to what? It is not clear state this and reference this.

This concluding sentence at L747-749 has been clarified to indicate that methanotrophs were not as widely detected as methanogens in the landfill cell and a smaller number of families and distinct populations (as measured by # of MAGs) were recovered for methanotrophs vs methanogens.

Lines 658-660: It is hard to compare between these because they are not on the same plot anywhere. Please put them together so that comparison can easily be made. Also, total abundances of lineages is preferable rather than just looking comparison of individual genomes as it makes the comparison even more difficult between groupings and different metabolism types.

See comments above regarding whole community and methane cycling guild summary data.

Lines 725-727: The mean coverages are a proxy and are valid for showing which are enriched, but these abundances are hard to read in the context of the community in Figure 2. Especially as the mean and median coverages of MAGs change between each site dependent on how much data is assembled into MAGs (also there is no data about how much GB of sequence data was produced and how much was assembled into MAGs), therefore it makes any sort of comparison between samples difficult. Why cannot a heat map of these community members relative to its abundance in the whole community be given? It would be more meaningful to put the methanogen/methanotroph community in context of the whole community not just what has assembled (which is likely to be variable between samples).

See comments above speaking to various data visualizations that summarize this information.

Reviewer #2 comments

Reviewer 2 requested additional geochemical summary data. We make note that many of Reviewer 2's comments lacked sufficient detail pertaining to which additional data interpretations should be included. The lack of detail makes it challenging to fully address. Please see our response to Reviewer 2's comments below.

The present research investigates the methane cycle with respect to microbial community population. The introduction provides sufficient background information. However, could consider reducing the length of the text in lines 84-100 and lines 84-91 may precede the last paragraph.

This paragraph has been streamlined in L90-100 and the topic sentence has been moved to the final paragraph of the introduction in L113.

Lines 101-111 are also very general and not provide any information on the subject, whereas the physicochemical characteristics are overlooked.

Additional emphasis has been placed on connecting methane cycling to physicochemical conditions to better understand which pathways support methane production and which niches can support different methanotrophic lifestyles in L106-109.

Lines 125-153 are a part of the methodology, where additional information on the landfill characteristics should be included (amount of waste deposited, climatic conditions, volume of leachate produced etc).

This content has been moved to a Site Description section L780-810 in the Methods that refers to the summary information requested in **Table S1**.

Lines 138-139, please be more specific when referring to active filling periods, as well as the condition of the landfills during the sampling.

Active landfilling periods are indicated in the subtitles for each geochemical section and are also available for quick reference in the new **Table S1**. We already include data about the landfill cells actively receiving waste and whether leachate recirculation was occurring. Outside of these conditions for the landfill cells, it is unclear what additional content could be included that would add to the qualitative site description.

The authors did not consider the value of the simple BOD/COD ratio, as in many cases researchers characterize a leachate sample based on the age on the landfill and not on the actual data. Authors should include a plain table containing the physicochemical characteristics of the raw samples and the respective measurements at the sampling day.

Historical data summarized in **Fig S1** for leachate geochemistry has been provided in **Table S1** for the sampling day 2019-02-12, which was one week prior to our sampling date. BOD/COD ratios for that day has also been provided and are referenced where appropriate in our interpretation of the geochemical data.

The ammonium nitrogen content and the electrical conductivity are overlooked. Alkalinity is between VFAs. All these crucial measurements are lost in supplementary material graphs.

We constrained our analyses to focus on carbon given the length we anticipated for this manuscript based on the MAG data. The geochemical records provided distinguish between total alkalinity and bicarbonate as separate variables, which is why we chose to present bicarbonate alongside other potential substrates for methanogenesis. Ammonia was originally plotted but did not fit into a focused narrative examining methane cycling guild physiology. Without additional details on this comment, we are unsure as to how the reviewer would like these variables added into the manuscript.

Did authors extract DNA only once? How many replicate samples were analysed from each site?

Yes, obtaining true replicate samples from the study site for that sampling day is not possible currently.

Fig 1B does not clearly present sample area D1, whereas oxygen presence in A and B is not adequately discussed and the pH.

We suspect the reviewer is referring to differences in the gas flaring data and aqueous geochemistry data provided in the supplemental information. This discrepancy in the naming originates from the information provided by site owners with respect to where the gas flaring vents used for gas sampling are located. We have attempted to reconcile the location of these vents and their connections as best possible in the manuscript. No notable differences in pH at the time of sampling were observed across the landfill cells, which all displayed circumneutral pH (summarized in **Table S1**). We think the potential infiltration of oxygen is discussed adequately in the context of physiological controls on methane cycling in the manuscript. Without further detail on this reviewer comment, it is difficult to understand what additional interpretation could be provided to improve the manuscript.

Line 248, please correct ‘‘Halobacteriota’’ (Rinke and co-workers, 2021, A standardized archaeal taxonomy for the Genome Taxonomy Database’).

We used the nomenclature associated with the GTDB release used for taxonomic classification (r89). We would prefer to maintain the taxonomic classifications associated with that release in the data presented for reproducibility. We have made note of the updated name and the associated GTDB release upon the first mention of the *Halobacterota* in the whole community analyses that has been added to the manuscript at L223 and the introduction to the methanogenic section at L285.

Could authors explain the sharp decrease of pH in cell E?

We attempt to explain this in L168 where a sharp decrease in bicarbonate and likely buffering capacity occurred. Site owners were unable to provide qualitative data as to whether highly acidic waste that could result in this depletion occurred that day.

Please include the main findings of the metagenomic analysis, even at family level, in a comparative table.

A new **Fig 2** that summarized beta diversity analyses has been provided alongside the interpretation of these results. Summary heatmaps at the phylum and family level for the whole community have also been provided in **Fig S3** and **Fig S4** alongside a dedicated section providing an overview of the whole microbial community and dominant taxa.

The dominance and relative abundance of each phylum and family should be clearly stated, like in line 470 and in methanotrophy results, instead of just the classification.

See comment above on data summarizing dominant phyla and families across the landfill. We have also included a summary figure for methane cycling guilds (**Fig S5**) to contextualize their overall relative abundance compared to the rest of the community. We have substituted coverage data for relative abundance data throughout our discussion surrounding the methane cycling communities and refer to the most abundant MAG at each landfill cell throughout the manuscript.

Why the methanol dehydrogenase gene (mx_aF) was not included in the research.

Examining methylotrophic metabolism, which is not necessarily methanotrophic, fell outside of the scope of this methane-centric manuscript.

Lines 664-648 are too general and not in compliance with lines 540-544 where reference is made regarding the presence of Methanoperedenaceae MAGs.

We think the level at which the information surrounding redox adaptations has been synthesized to an appropriate level. Some clarification around this comment connecting with our detection of the *Methanoperedenaceae* is needed given that the paragraph following the one associated with this comment anaerobic and intra-aerobic methanotrophy starting at the new L754.

The methodology of physicochemical analyses is absent.

These records were provided by site owners. We have indicated the consulting company that carried out these analyses and indicated that they used standardized methods for landfill leachate monitoring. We think that adding additional methodological details will not improve the manuscript, which in its current form, must be streamlined in line with reviewer comments.

The research is very interesting, but most findings are scattered in numerous supplementary figures and tables, which do not help the clarity of the presentation.

We have attempted to streamline the manuscript as much as possible while also including the data analyses requested. We have streamlined the geochemical interpretations to focus on gas flaring data, which largely informed our classification of the landfill cells within the lifecycle model and make brief mentions of aqueous geochemistry and carbon substrates as needed. We have also attempted to streamline our interpretation surrounding the methane cycling community and include overview sections of the whole-community and methane cycling guilds to facilitate microbiological data interpretation.

Reviewers' Comments:

Reviewer #1:

Remarks to the Author:

Thank you to the authors for answering my queries from the review. I apologise for the length of time to get back to the author rebuttal.

The only issue I have is outlined below:

I agree with the authors that presenting objective material from metabolic annotation software is a good view to have, but also at the same time I don't want information being perpetuated through the literature about lineages having a certain type of metabolism based on the presence of a gene (acetyl-CoA synthetase) for which downstream metabolic processes are significantly different (acetoclastic methanogenesis vs acetate assimilation).

To this end I looked through the literature that the reviewers provided for the sentence on lines 516-518 for evidence of acetoclastic methanogenesis from the *Methanoculleus* and *Methanobrevibacter* species as I was very interested as this would change how I view the substrate utilization by these lineages. However, in the context of the sentence I feel that there is no conclusive evidence for happening in these study or review articles. For example:

Reference 53 is a mixed culture, and it cannot be conclusively concluded that methanogenesis from *Methanoculleus* is coming from acetate, despite SIP labelled acetate being present. In theory, acetate could be oxidised to CO₂ by non-*Methanoculleus* organisms and then methane formed via hydrogenotrophic pathway by *Methanoculleus* species.

Reference 54 references to *Methanotrichaceae* for which it does use the acetoclastic methanogenesis pathway but isn't mentioned in this sentence and doesn't validate the *Methanoculleus* or *Methanobacteriaceae* utilising acetoclastic methanogenesis pathway.

Reference 56 talks about *Methanobacteriaceae* requiring acetate as a carbon source (consistent with the sentence added at Line 516) but doesn't mention about them forming methane from acetate.

Reference 57 talks about a mixed culture of *Methanotrichaceae* and *Methanobacteriaceae* being responsible for methane. Based on the evidence they it cannot be said that the *Methanobacteriaceae* are acetoclastic methanogens based on the presence of *Methanotrichaceae*, it is likely they are assimilating acetate as a carbon source.

Reference 58 doesn't provide evidence for acetoclastic methanogenesis by *Methanoculleus*, suggest there may only be syntrophic mechanism for acetate being oxidised to H₂/CO₂ that is then utilised by *Methanoculleus* species.

Reference 59 references other primary literature which also suggest there is no methanogenesis from acetate in *Methanoculleus* isolates.

Reference 60 the machinery identified here suggests that this *Methanoculleus* methanogen assimilates acetate, rather than is able to produce methane from acetate. There is no evidence for acetoclastic methanogenesis.

For the comment from the author rebuttal, 'We think it is best to present this data objectively based on our classification framework, highlighting discrepancies in our data compared to the current literature.' I think rather than pointing out there are discrepancies in the data and literature, I think it should be pointed out there is a discrepancy in the software with the literature. Alternatively, one small experiment would show this, can DRAM be run on existing *Methanoculleus* isolate genomes to prove that they don't have the ability for acetoclastic methanogenesis? As they are in pure culture and aren't able to utilise acetate then they won't have the gene(s) for acetoclastic methanogenesis. I can live with either of these compromises. Also, sentence L516-518 needs to be reworked to reflect that the literature provided here doesn't quite confirm all of the

information in this sentence.

To this end about incorrect interpretations of methanogenic metabolism types, for the query:

Reviewer: Line 397: Are these Methanosarcinaceae utilising H₂/CO₂ or do they just have the pathway? As such, seen in the *Methanosarcina barkeri*, has hydrogenotrophic pathway but doesn't utilise it for H₂/CO₂ reduction.

Authors: We can only discuss the presence of pathways in this manuscript. Presumably, these pathways are maintained because they are useful under specific conditions in the landfill that may not have been successfully reproduced in past work with *Methanosarcina barkeri*.

Yes, this pathway is maintained for specific reasons. Such as methylotrophic methanogenesis in *Methanosarcina* requires disproportionation of methanol via the reverse of the CO₂ reduction pathway. Hence while not utilising H₂/CO₂, they require for methylotrophic methanogenesis. I would like this query revisited by the authors.

Reviewer #2:

Remarks to the Author:

The authors addressed most concerns regarding the manuscript, which could be accepted for publishing.

We thank the reviewer for their suggestions, which we have taken seriously, and addressed as described below.

Reviewer #1 (Remarks to the Author):

Thank you to the authors for answering my queries from the review. I apologies for the length of time to get back to the author rebuttal.

The only issue I have is outlined below:

I agree with the authors that presenting objective material from metabolic annotation software is a good view to have, but also at the same time I don't want information being perpetuated through the literature about lineages having a certain type of metabolism based on the presence of a gene (acetyl-CoA synthetase) for which downstream metabolic processes are significantly different (acetoclastic methanogenesis vs acetate assimilation).

To this end I looked through the literature that the reviewers provided for the sentence on lines 516-518 for evidence of acetoclastic methanogenesis from the Methanoculleus and Methanobrevibacter species as I was very interested as this would change how I view the substrate utilization by these lineages. However, in the context of the sentence I feel that there is no conclusive evidence for happening in these study or review articles. For example:

Reference 53 is a mixed culture, and it cannot be conclusively concluded that methanogenesis from Methanoculleus is coming from acetate, despite SIP labelled acetate being present. In theory, acetate could be oxidised to CO₂ by non-Methanoculleus organisms and then methane formed via hydrogenotrophic pathway by Methanoculleus species.

Reference 54 references to Methanotrichaceae for which it does use the acetoclastic methanogenesis pathway but isn't mentioned in this sentence and doesn't validate the Methanoculleus or Methanobacteriaceae utilising acetoclastic methanogenesis pathway.

Reference 56 talks about Methanobacteriaceae requiring acetate as a carbon source (consistent with the sentence added at Line 516) but doesn't mention about them forming methane from acetate.

Reference 57 talks about a mixed culture of Methanotrichaceae and Methanobacteriaceae being responsible for methane. Based the evidence they it cannot be said that the Methanobacteriaceae are acetoclastic methanogens based on the presence of Methanotrichaceae, it is likely they are assimilating acetate as a carbon source.

Reference 58 doesn't provide evidence for acetoclastic methanogenesis by Methanoculleus, suggest thee yar only syntrophic mechanism for acetate being oxidise to H₂/CO₂ that is then utilised by Methanoculleus species.

Reference 59 references other primary literature which also suggest there is no methanogenesis from acetate in Methanoculleus isolates.

Reference 60 the machinery identified here suggests that this Methanoculleus methanogen assimilates acetate, rather than is able to produce methane from acetate. There is no evidence for acetoclastic methanogenesis.

For the comment from the author rebuttal, 'We think it is best to present this data objectively based on our classification framework, highlighting discrepancies in our data compared to the current literature.' I think rather than pointing out there are discrepancies in the data and literature, I think it should be pointed out there is a discrepancy in the software with the literature. Alternatively, one small experiment would show this, can DRAM be run on existing Methanoculleus isolate genomes to prove that they don't have the ability for acetoclastic methanogenesis? As they are in pure culture and aren't able to utilise acetate then they won't have the gene(s) for acetoclastic methanogenesis. I can live with either of these compromises. Also, sentence L516-518 needs to be reworked to reflect that the literature provided here doesn't quite confirm all of the information in this sentence.

To this end about incorrect interpretations of methanogenic metabolism types, for the query:

Reviewer: Line 397: Are these Methanosarcinaceae utilising H₂/CO₂ or do they just have the pathway? As such, seen in the Methanosarcina barkeri, has hydrogenotrophic pathway but doesn't utilise it for H₂/CO₂ reduction.

Authors: We can only discuss the presence of pathways in this manuscript. Presumably, these pathways are maintained because they are useful under specific conditions in the landfill that may not have been successfully reproduced in past work with Methanosarcina barkeri.

Yes, this pathway is maintained for specific reasons. Such as methylotrophic methanogenesis in Methanosarcina requires disproportionation of methanol via the reverse of the CO₂ reduction pathway. Hence while not utilising H₂/CO₂, they require for methylotrophic methanogenesis. I would like this query revisited by the authors.

Response to Reviewer 1: Following Reviewer 1's suggestion, we analyzed seven isolate genomes from the *Methanoculleus* genus using DRAM. The product file output by DRAM indicated that each genome harboured the *mcrA* gene, as expected. All genomes also carried the genes coding the acetyl-CoA synthetase (noted as Acetate pt 1 in DRAM's output) and full hydrogenotrophic methanogenesis pathways. These observations suggest that each strain has the metabolic potential convert acetate to acetyl-CoA. The classification rules we employed in our study required that genomes identified as putative acetoclastic methanogens have a 50% or higher completion for the carbonic anhydrase/acetyl-CoA synthase (CODH/ACS) pathway. 3/7 of the isolate genomes analyzed had 50% completion for the CODH/ACS pathway (i.e., *Methanoculleus bourgensis* MS2, *Methanoculleus chikugoensis* MG62, and *Methanoculleus taiwanensis* CYW4), which would classify them as putative acetoclastic methanogens using our system.

From physiological studies on *M. bourgensis* MS2, this strain is considered a hydrogenotrophic methanogen that requires acetate as a carbon source for growth. Previous work on this strain detected the presence of the acetyl-CoA synthetase gene^{1,2}, demonstrating that DRAM is arriving at the same annotation result observed by others. Previous work also suggests that the occurrence of the acetyl-CoA synthetase pathway enables *M. bourgensis* MS2 to metabolize

acetate through the carbonyl branch of the Wood-Ljungdahl pathway². Our own annotation results suggest only 43% of the genes required for the Wood-Ljungdahl pathway are present in this strain. The genomic characterization carried out in previously published studies aligns with the original work isolating this strain in 1986, where 1 g per litre (~ 16 mM) of acetate was supplied to cells as a required growth substrate, which was not used for methanogenesis³.

From studies on *M. chikugoensis* MG62, the isolation of this strain from paddy soil demonstrated it could use a variety of organic substrates to produce methane, but not acetate⁴. ¹³C-labelling experiments in dairy and swine manures suggest that populations related to unidentified species in the *Methanoculleus* genus and *M. chikugoensis* assimilated acetate-derived carbon into DNA⁵. As noted by Reviewer 1, this observation wouldn't preclude the assimilation of ¹³C-labelled acetate directly as a carbon source or the assimilation of ¹³C-carbon dioxide generated from the oxidation of the ¹³C-labelled acetate through syntrophy.

From studies on *M. taiwanensis* CYW4, a comparative genomic study indicating genes coding for the CODH/ACS pathway were annotated in *Methanoculleus* members such as *M. bourgensis* MS2, but not *M. taiwanensis* CYW4⁶. This observation runs counter to our annotations with DRAM, suggesting different bioinformatics pipelines and database updates can lead to different annotation results. Although the authors of the study highlighted the importance of genes coding the CODH/ACS pathway in acetoclastic methanogenesis, the importance of this pathway in the context of using carbon monoxide as a methanogenic substrate is emphasized rather than its role in acetoclastic methanogenesis⁶. The original paper detailing the isolation of *M. taiwanensis* CYW4 tested whether 50 mM acetate could support catabolism, and added 20 mM acetate and 100 mM formate as stimulatory growth substrates⁷. That study concluded acetate was not used as a methanogenic substrate and includes a table reinforcing that *M. bourgensis* MS2 and *M. chikugoensis* MG62 require acetate as a growth substrate⁷.

By cross-referencing the *Methanoculleus* isolate strains annotated using DRAM with published physiological experiments, we have highlighted one of the limitations of using a purely metagenomic approach to resolve downstream metabolic processes contributing to methane production. Although we designed a classification system to capture diverse putative acetoclastic methanogens bearing in mind some genome bins would have <100% completion, our approach is unable to discern whether the same machinery involved in acetate metabolism is coupled to methanogenesis or carbon assimilation.

We revisited the summary interpretation provided by Reviewer 1 for the citations supporting our comment on the conflicting evidence surrounding acetoclastic methanogenesis in the families *Methanotrichaceae*, *Methanobacteriaceae*, and *Methanocullaceae*. We acknowledge that including all these references in a single sentence was confusing and did not speak to the specific discrepancies we wanted to highlight for each family. The *Methanotrichaceae* are generally agreed upon to be acetoclastic, whereas the *Methanobacteriaceae* and *Methanocullaceae* are not. We have reframed the context in which we use these citations to indicate that many studies have examined the physiology of acetate metabolism in these families (the *Methanocullaceae* in particular) and shown that acetate is often a required carbon source but not a confirmed substrate for methane production. These edits have been implemented throughout the manuscript at L316, L469-481, L484-486, and L516-520.

L316: According to classification rules used in this study (see Methods),

L469-481: We note MAGs from the *Methanocullaceae* family were mixed as to their classification as hydrogenotrophic or acetoclastic methanogens across multiple landfill cells based on the rules we used to classify putative methanogenic metabolisms. Physiological studies on isolates from the *Methanoculleus*, the only genus from the *Methanocullaceae* recovered from the landfill (Supplementary Data 1 on the Open Science Framework (OSF) at DOI 10.17605/OSF.IO/6X5ZC³⁷), have shown that acetate is a required carbon source that does not serve as a substrate for methane production under the conditions tested^{51–57}. Members from the *Methanobacteriaceae* were subject to a similarly mixed classification in our study despite previous work showing that acetate is not a methanogenic substrate (compiled in⁵⁸).

L516-520: Members of the *Methanotrichaceae* are considered acetoclastic methanogens in line with our metabolic modeling⁶² and we have already noted the potential limitations of members of the *Methanocullaceae* and *Methanobacteriaceae* as acetoclastic methanogens in the absence of physiological evidence. Regardless of which substrates support methane production, these families were repeatedly detected in newer waste but also waste that was over 20 years old...

Reviewer 1 also indicated that original L397 in the manuscript should be revisited to distinguish between H₂/CO₂ machinery involved in hydrogenotrophic methanogenesis vs the disproportionation of methanol via the reverse of the CO₂ reduction steps. We added an additional sentence and references^{8,9} to our manuscript at line 480-482 to address this point:

“Similar limitations must be considered for methanogens that carry genes for hydrogenotrophic pathways to support the disproportionation of methyl-bearing substrates used for methane production^{60,61}.”

References

1. Maus, I. *et al.* Complete genome sequence of the hydrogenotrophic, methanogenic archaeon *Methanoculleus bourgensis* strain MS2^T, isolated from a sewage sludge digester. *J. Bacteriol.* **194**, 5487–5488 (2012).
2. Maus, I. *et al.* Insights into the annotated genome sequence of *Methanoculleus bourgensis* MS2^T, related to dominant methanogens in biogas-producing plants. *J. Biotechnol.* **201**, 43–53 (2015).
3. Ollivier, B. M., Mah, R. A., Garcia, J. L. & Boone, D. R. Isolation and characterization of *Methanogenium bourgense* sp. nov. *Int. J. Syst. Evol. Microbiol.* **36**, 297–301 (1986).
4. Dianou, D. *et al.* *Methanoculleus chikugoensis* sp. nov., a novel methanogenic archaeon isolated from paddy field soil in Japan, and DNA-DNA hybridization among *Methanoculleus* species. *Int. J. Syst. Evol. Microbiol.* **51**, 1663–1669 (2001).
5. Barret, M. *et al.* Phylogenetic identification of methanogens assimilating acetate-derived carbon in dairy and swine manures. *Syst. Appl. Microbiol.* **38**, 56–66 (2015).
6. Chen, S.-C. *et al.* Comparative genomic analyses reveal trehalose synthase genes as the signature in genus *Methanoculleus*. *Marine genomics.* **47**, 100673 (2019).
7. Weng, C.-Y. *et al.* *Methanoculleus taiwanensis* sp nov., a methanogen isolated from deep marine sediment at the deformation front area near Taiwan. *Int. J. Syst. Evol. Microbiol.* **65**, 1044–1049 (2015).
8. Meuer, J. *et al.* Genetic analysis of the archaeon *Methanosarcina barkeri* Fusaro reveals a central role for Ech hydrogenase and ferredoxin in methanogenesis and carbon fixation. *Proceedings of the National Academy of Sciences of the United States of America.* **99**,

- 5632–5637 (2002).
9. Welander, P. V, Metcalf, W. W., Welander, P. V & Metcalf, W. W. Mutagenesis of the C1 oxidation pathway in *Methanosarcina barkeri*: New insights into the Mtr/Mer bypass pathway. *Journal of bacteriology*. **190**, 1928–1936 (2008).